# Removing Concepts from Text-to-Image Models with Only Negative Samples

**Hanwen Liu, Yadong Mu**[*]
Peking University
`hanwenliu@msn.com, myd@pku.edu.cn`

## Abstract

This work introduces `Clipout`, a method for removing a target concept in pre-trained text-to-image models. By randomly clipping units from the learned data embedding and using a contrastive objective, models are encouraged to differentiate these clipped embedding vectors. Our goal is to remove private, copyrighted, inaccurate, or harmful concepts from trained models without the need for retraining. This is achieved by considering only negative samples and generating them in a bootstrapping-like manner, requiring minimal prior knowledge. Additionally, theoretical analyses are provided to further understand our proposed `Clipout`. Extensive experiments on text-to-image show that `Clipout` is simple yet highly effective and efficient compared with previous state-of-the-art approaches.

## 1 Introduction

Modern deep learning research is driven by scale. Powered by tremendous resources in computing and data, large-scale models make it possible to connect the vision and language domains using a single architecture (Radford et al., 2021; Dosovitskiy et al., 2021), *i.e.*, vision-language models (VLMs). Modern text-to-image models (Ramesh et al., 2021; Yasunaga et al., 2023), such as the famed generative model Stable Diffusion (Rombach et al., 2022), are capable of high-quality and diverse image synthesis based on a text prompt. For image-to-text tasks (Stefanini et al., 2023), based on the large-scale data of image-text pairs, VLMs (Li et al., 2022b) can describe visual concepts of an image in natural language. These studies continue to evolve and scale, leading to various cases in real-world scenarios (Li et al., 2022a; Xu et al., 2025).

While the scaled-up models and datasets have made monumental progress, there is an increasing risk that the learned concepts in the trained models may be used maliciously. By leveraging personalized text-to-image techniques (Ruiz et al., 2023; Gal et al., 2023), personal portraits can be forged into deepfakes (Shiohara and Yamasaki, 2022; Chen et al., 2021; Yu et al., 2021) by text-to-image models, and aided by these models, the art style of paintings can also be easily imitated without authorization (Liang et al., 2023). Besides, since it is nearly impossible to carefully audit large-scale datasets, models trained on these unfiltered data may present biased and harmful behaviors (Birhane et al., 2021). These misuses of VLMs, especially text-to-image models, will severely violate the rights of privacy and copyright, making the development of the scaling model go against its original intention.

A feasible solution to these issues is machine unlearning (Xu et al., 2024), which grants people *the right to be forgotten*. The primary challenge behind machine unlearning is how to *efficiently* and *precisely* unlearn the target concept from trained models. Early research work (Bourtoule et al., 2021; Yan et al., 2022) focused on partitioning the dataset and model into several shards and sub-models, and re-trained the sub-model corresponding to the target concept. However, these methods require re-design of the model architecture, and cannot be applied to modern scaled-up vision-language

---

[*]Corresponding author.

39th Conference on Neural Information Processing Systems (NeurIPS 2025).

models. Recent efforts have been made toward unlearning copyright or harmful data in trained text-to-image models (Gandikota et al., 2023; Kumari et al., 2023). These methods operate by fine-tuning the trained diffusion model in an end-to-end fashion, and there is a trade-off between efficiency, effectiveness, and the side effects introduced by the change of model parameters. As modern deep neural networks, though large in scale, are built upon different foundational artifacts (Radford et al., 2021), this paper aims to investigate the question: *Can we remove particular concepts in trained text-to-image VLMs, by only unlearning the foundational artifacts that build up these models?*

To unlearn text-to-image models at scale, this paper introduces a conceptually simple method, `Clipout`, by focusing on the embedding of the learned concepts. Unlike previous methods that treat the vision-language model as a whole, `Clipout` decouples the complex text-to-image system into several distinct artifacts (*e.g.*, the CLIP encoder (Radford et al., 2021) used in Stable Diffusion (Rombach et al., 2022)), and only unlearns the artifact that is responsible for generating the concept embedding. Inspired by the high efficiency and strong generalization of contrastive learning, in this paper, we explore the opposite direction and propose contrastive *unlearning*, which aims at unlearning a particular concept by introducing a novel contrastive objective. Based on the contrastive unlearning paradigm, `Clipout` can unlearn target concepts in a more efficient and precise way.

**Contributions.**    a) We propose the first machine unlearning method based on a contrastive objective with only negative samples considered. b) Utilizing contrastive learning theories, we provide necessary theoretical analyses on the effectiveness of the proposed unlearning method. c) Through extensive experiments, our proposed method not only exceeds previous state-of-the-art methods in performance but also has significant superiority in efficiency.

## 2   Background

Contrastive objectives (Zbontar et al., 2021; Huang et al., 2023b) have been widely adopted in vision-language pre-training tasks, and many VLMs use image-text pairs to calculate these losses. InfoNCE (van den Oord et al., 2018) is one of the most canonical contrastive losses. Radford et al. (2021) proposed CLIP to bridge vision and language domains by learning image and text encoders based on InfoNCE. Following the idea of CLIP, a series of methods have been proposed (Jia et al., 2021; Li et al., 2021). For image generation tasks, Rombach et al. (2022) proposed Stable Diffusion, which runs the diffusion process in the latent space to generate photo-realistic images. Stable Diffusion uses a fixed CLIP encoder to condition the images on the prompt.

Machine unlearning (Bourtoule et al., 2021; Ullah and Arora, 2023; Tarun et al., 2023) allows trained models to discard specific information and maintain privacy by erasing the influence specific data points have on the models. Different from *unlearnable data* (Huang et al., 2021; Liang et al., 2023) that safeguards personal privacy by adding the protective noise into data ahead of time, machine unlearning aims to modify a trained model to delete certain data points. While most methods focus on small-scale or discriminative tasks, some recent studies have started to pay attention to VLM scenarios, especially in unlearning Stable Diffusion (Heng and Soh, 2023; Gandikota et al., 2024). Kumari et al. (2023) proposed to ablate concepts in Stable Diffusion by overwriting them with a general category anchor concept. Another concept-erasing method has been proposed by Gandikota et al. (2023), which shares a similar motivation behind compositional energy-based models (Du et al., 2020). Additionally, Schramowski et al. (2023) proposed an inference-time method, namely Safe Latent Diffusion, to mitigate inappropriate concepts. Amara et al. (2025) discussed the ripple effects that may extend beyond the target concepts to erase. Kurmanji et al. (2023) proposed to use a contrastive-like loss to unlearn small nets for classification tasks, while we proposed to unlearn large-scale VLMs for image generation tasks. Generally, the text-to-image models intrinsically contain the text encoder for prompt conditioning. By unlearning the text encoder, our method can precisely remove the undesired concepts in the model, as the model cannot find the correct condition.

## 3   Contrastive Unlearning

We propose `Clipout` to unlearn private, copyrighted, or inappropriate concepts in text-to-image VLMs. Based on the hypothesis of *latent class* (Wang and Isola, 2020; Huang et al., 2023a), the term *concept* in this paper refers to a distribution that samples from the same concept share the same semantic meaning, and it is described by the prompt in text-to-image generation. We only unlearn a

(1) Unlearning the encoder *w.r.t.* a target data embedding

(2) Using the unlearned encoder

Figure 1: The pipeline of `Clipout`. Given a prompt that describes the concept to unlearn, `Clipout` first generates variants of the embedding by randomly masking the vector and then uses the contrastive objective to optimize the encoder. These masked vectors, *e.g.*, $z_i z_j$, serve as negative samples in unlearning. The (red) positive samples, *e.g.*, $z_i z_i$, are omitted in the optimization. The encoder is updated by maximizing the dissimilarity between these masked variants. After unlearning, the updated encoder replaces the original one, so the model can no longer generate the removed concept.

part of VLMs, namely the encoder that produces the embedding to generate conditions for other parts, and leave other parts fixed. Next, we will introduce the objective used in `Clipout` with theoretical analyses. The following assumptions are made for simplicity. Proofs are present in the *appendix*.

**Assumption 3.1.** For any neural net $f_\theta(\cdot)$ parameterized by $\theta$ from a function class $\mathcal{F}$, $f_\theta(\cdot)$ is continuous. Likewise, we assume the similarity function $\text{sim}(\cdot)$ is also continuous and bounded.

**Assumption 3.2.** In contrastive learning, the embedding $z'$ and $z$ from $f_\theta(\cdot)$ share the same *latent class* if $||z - z'||^2$ is not too large, and $z'$ with $z$ can form a positive pair *w.r.t.* this latent class based on empirical studies (Gao et al., 2021). By default, $|| \cdot ||$ denotes Frobenius or $l_2$-norm.

### 3.1 Negative-Only Contrastive Loss

In text-to-image models, a concept is the semantic idea or category triggered by a text prompt, which, in the context of representation learning, corresponds to a semantic distribution in the embedding space that ensures all conditioned generated samples share the same underlying semantics.

The gist of `Clipout` is to make the concept embedding, which is used to represent the conditions, unable to identify itself. Given a trained encoder $f_\theta(\cdot)$ and the target embedding $f_\theta(x) = z \in \mathbb{R}^d$ for an input data point $x$, we randomly clip out some units from the embedding and set them to zero. Assume the mask $m \in \{0, 1\}^d$, the contrastive loss, which is inspired by InfoNCE, yields:

$$\ell_\theta = \frac{1}{N} \sum_{i=1}^{N} - \log \frac{e^{\text{sim}(z \odot m_i, z \odot m_i)/\tau}}{\sum_{j=1}^{N} e^{\text{sim}(z \odot m_i, z \odot m_j)/\tau}}, \tag{1}$$

where $N$ is the batch size, $\tau > 0$ is the temperature parameter, and $\text{sim}(\cdot)$ is the similarity function. Assume $\text{sim}(z \odot m_i, z \odot m_i) = 1$, we have:

$$\ell_\theta = \frac{1}{N} \sum_{i=1}^{N} \log \sum_{j=1}^{N} e^{\text{sim}(z \odot m_i, z \odot m_j)/\tau} - \frac{1}{\tau}. \tag{2}$$

The element in $m$ is *i.i.d.* drawn from Bernoulli($p$), and $m$ can be viewed as the vector of $d$ Bernoulli trials. We term $p$ as the *clipout rate* analogous to the dropout rate (Srivastava et al., 2014). In a single training epoch, $\ell_\theta$ can be regarded as the empirical version of the population loss:

$$\ell_\theta = \mathbb{E}_{m_i} \Big[ \log \sum_{j=1}^{N} e^{\text{sim}(z \odot m_i, z \odot m_j)/\tau} \Big] - \frac{1}{\tau}, \tag{3}$$

where $m_i$ and $m_j$ are two independent random variables. According to Jensen's inequality, we have:

$$\ell_\theta \geq \frac{1}{\tau} \mathbb{E}_{m_i, m_j}[\text{sim}(z \odot m_i, z \odot m_j)] + \log N - \frac{1}{\tau}, \tag{4}$$

$$\ell_\theta \leq \log \mathbb{E}_{m_i, m_j}[e^{\text{sim}(z \odot m_i, z \odot m_j)/\tau}] + \log N - \frac{1}{\tau}.$$

As our goal is to minimize $\ell_\theta$, Eq. (4) largely reduces to:

$$\min_\theta \mathbb{E}_{m_i, m_j} \big[ \text{sim}(z \odot m_i, z \odot m_j) \big]. \tag{5}$$

By generating multiple variants of $z$, we treat these clipped embedding vectors $z \odot m$ as *negative samples*, and make them *dissimilar* from themselves. Since we do not use positive samples (*i.e.*, samples that should be gathered together) and only focus on negative samples in the unlearning objective, it is implied that the temperature $\tau$ is not as important as in conventional contrastive learning methods (Chen et al., 2020; He et al., 2020). The pipeline of our method is presented in Figure 1. Given a target prompt, it requires independent optimization for unlearning, and no extra corpus is needed for fine-tuning.

### 3.2 On the Effectiveness of Unlearning

An empirical justification for randomly masking the embedding vectors is that these masked vectors serve as the *positive samples* for the input data point in contrastive learning (Gao et al., 2021; Xu et al., 2023). As we treat masked vectors as negative samples, we anticipate effective analyses with theoretical guarantees. In this part, we first analyze the convergence of our proposed *negative-only contrastive loss* in Eq. (1) and then demonstrate why `Clipout` is effective, from the perspectives of *alignment* and *uniformity* (Huang et al., 2023a; Wang and Isola, 2020).

**Convergence.** From an intuitive perspective, Eq. (5) demonstrates that by optimizing $\ell_\theta$, the representation of $z$ for a particular concept can be destroyed and it no longer represents its original semantic meaning. Besides, we can obtain the convergence guarantee of $\ell_\theta$ as in Proposition 3.3.

**Proposition 3.3** (Convergence). *Given the temperature $\tau > 0$ and the batch size $N$, if $\{m_i\}_{i=1}^N$ are i.i.d. and $sim(z \odot m_i, z \odot m_i) = 1$ for all $i$, $\ell_\theta$ in Eq. (1) converges to:*

$$\lim_{N \to \infty} (\ell_\theta - \log N) = \mathbb{E}_{m_i}\big[\log \mathbb{E}_{m_j}[e^{sim(z \odot m_i, z \odot m_j)/\tau}]\big] - \frac{1}{\tau}. \tag{6}$$

Compared with theoretical results in previous work (Huang et al., 2023a; Wang and Isola, 2020), Proposition 3.3 implies that our loss converges to a point where the distribution of $m$ matters, as there is no positive sample in our loss. Since we only consider variants of a single target embedding, the batch size $N$ (*i.e.*, the number of variants) can be (roughly) interpreted as the number of iterations in the optimization. Proposition 3.3 has ensured that the loss considered in `Clipout` tends to be convergent after masking the target embedding infinite times, and we now demonstrate the speed of convergence as in Corollary 3.5.

**Lemma 3.4** (The upper bound of $\mathbb{E}\big[|\frac{1}{N}\sum_{i=1}^N x_i|\big]$ (Wang and Isola, 2020)). *Given i.i.d. random variable $x_i$ with bounded support $\subset [-a, a]$, zero mean and $\sigma_x^2 \leq a^2$ variance, the expected value $\mathbb{E}\big[|\frac{1}{N}\sum_{i=1}^N x_i|\big]$ is bounded by $\mathcal{O}(\frac{1}{\sqrt{N}})$.*

**Corollary 3.5** (Convergence speed). *The error term decays in $\mathcal{O}(\frac{1}{\sqrt{N}})$, w.r.t. the convergence speed of the limit in Proposition 3.3 according to Lemma 3.4.*

Corollary 3.5 encourages more negative samples in the optimization. As the dimension of vectors is usually compact, it is feasible to involve a large batch size in practice.

**Unlearning the target concept.** To investigate whether `Clipout` can unlearn the target concept while limiting the bad impact on other concepts, we try to analyze the problem from the perspective of *alignment* and *uniformity*, as in Definition 3.6.

**Definition 3.6** ($\alpha$-alignment and $\beta$-uniformity (Huang et al., 2023a; Wang and Isola, 2020)). In the contrastive objective, *alignment* denotes positive pairs should be mapped to nearby features, and *uniformity* represents feature vectors should be uniformly distributed on the unit hypersphere to preserve the data information. Assume that features are normalized, *i.e.*, $||f(\cdot)||^2 = 1$, these are defined as follows:

$$\ell_{\text{align}}(f; \alpha) = \mathbb{E}_{x, x^+}\big[||f(x) - f(x^+)||^\alpha\big], \tag{7}$$

$$\ell_{\text{uniform}}(f; \beta) = \log \mathbb{E}_{x, x'}\big[e^{-\beta||f(x) - f(x')||^2}\big], \tag{8}$$

where $\alpha > 0, \beta > 0$, $x$ with $x^+$ are positive samples, and $x$ with $x'$ are *i.i.d.* drawn from the input data distribution.

---

**Algorithm 1** Unlearning the encoder via `Clipout`

---

**Parameter**: prompt $x$ and encoder $f_\theta(\cdot)$

1: **for** $it = 1, iteration$ **do**
2:     Compute the data embedding $z = f_\theta(x)$;
3:     Sample $minibatch$ of $N$ masked samples from $z$;
4:     Compute the contrastive loss $\ell_\theta$ *w.r.t.* Eq. (1);
5:     Update $\theta$ by descending the gradients: $\nabla_\theta \ell_\theta$;
6: **end for**
7: return unlearned encoder parameters $\theta^*$;

---

*Remark* 3.7. Given a pre-trained encoder, alignment and uniformity can be used to measure the unlearning performance. To unlearn a target concept, the unlearned encoder is supposed to have poor alignment *w.r.t.* the target, as poor alignment means a deteriorated representation of the target. Meanwhile, the unlearned encoder should have good uniformity for any pair sampled from the input distribution, because the maximal information in the feature space should be preserved to maintain the model performance on unrelated tasks *w.r.t.* other concepts.

Consider the cosine similarity in Eq. (5), we have:

$$\mathbb{E}_{m_i, m_j}\big[\text{sim}(z \odot m_i, z \odot m_j)\big] \tag{9}$$
$$= 1 - \frac{1}{2}\mathbb{E}_{m_i, m_j}\big[||\frac{z \odot m_i}{||z \odot m_i||} - \frac{z \odot m_j}{||z \odot m_j||}||^2\big].$$

To minimize the loss, Eq. (9) implies that it needs to maximize the angle between two randomly masked embedding vectors, regardless of the magnitudes. Previous kinds of literature (Gao et al., 2021; Xu et al., 2023) have empirically proved that masked variants can be viewed as positive samples. Consider $\alpha = 2$ in Eq.(7), we build the connection between the unlearning objective and the loss of alignment in Definition 3.6:

$$\mathbb{E}_{m_i, m_j}\big[\text{sim}(z \odot m_i, z \odot m_j)\big] = 1 - \frac{1}{2}\ell_{\text{align}}(f_\theta; 2). \tag{10}$$

As in Definition 3.6, to learn a good representation, conventional contrastive learning aims at minimizing *alignment* of positive samples, *i.e.*, gathering these samples with the same latent semantics closely in the feature space. On the contrary, Eq. (10) suggests our proposed unlearning objective intends to *maximize* the target's alignment loss by increasing the Euclidean distance between their unit vectors on the feature hypersphere. As a result, a pair of masked variants of $z$ tends to become two vectors in opposite directions. By making a learned representation scattered on the feature space, `Clipout` can effectively unlearn the target concept by undermining the latent representations.

**Influence on different concepts.**    Naively, one can always unlearn the target by adding noise to the model parameters. However, arbitrarily changing the parameters will make the model useless for other concepts. When maximizing the target alignment, the uniformity loss should not increase.

**Lemma 3.8** (Uniform). *For any $x$ i.i.d. drawn from the input distribution, if the distribution of $f_\theta(x)$ is the uniform distribution $\sigma_d$, $\theta$ forms the minimizer for $\ell_{uniform}(f_\theta; \beta)$.*

*Remark* 3.9. For the unlearned encoder, we hope that the unlearned encoder with changed parameters still preserves as much information as possible. This requires the loss to converge to a point where the unlearned encoder can roughly form a uniform distribution on the feature hypersphere.

As we have demonstrated that optimizing Eq. (1) helps to unlearn the target embedding, according to Lemma 3.8, we can alleviate side effects on other concepts if the normalized feature distribution is the uniform distribution. Consider normalized vectors $u$ and $v$ where $u, v \in \mathbb{R}^d$, $||u|| = 1$ and $||v|| = 1$. Taking $u^\mathsf{T} v = \frac{z \odot m_i}{||z \odot m_i||}^\mathsf{T} \cdot \frac{z \odot m_j}{||z \odot m_j||}$, for Eq. (6) in Proposition 3.3 we have:

$$\mathbb{E}_{m_i}\big[\log \mathbb{E}_{m_j}[e^{\text{sim}(z \odot m_i, z \odot m_j)/\tau}]\big] \tag{11}$$
$$= \mathbb{E}_{m_i}\big[\log \mathbb{E}_{m_j}[e^{u^\mathsf{T} v/\tau}]\big].$$

Eq. (11) can be largely reduced to Eq. (12) in Lemma 3.10 with proper clipout rates, since we mask these vectors first and then normalize them. Note that Eq. (11) can only ensure a uniform distribution over the masked embedding vectors. To maintain the maximal information at the global level, we have to ensure the normalized feature vector $z/||z||$ is uniformly distributed on the unit hypersphere.

Table 1: Numerical results on VGGFace2 (Cao et al., 2018). The *sks* person introduced by different personalized methods (*e.g.*, DreamBooth (Ruiz et al., 2023)) is supposed to be unlearned, and similar concepts, such as *female* persons, ought to be kept. $\uparrow$ and $\downarrow$ indicate that the higher and lower values denote better performance, respectively. The prompts *a photo of sks person*, *a photo of female person*, and *a photo of male person* are used. Diff. represents the difference between the metrics of the target concept and the related concept. Values inside the bracket denote the difference compared with the results in the original pre-trained model.

| *Textual Inversion* | CLIP Score | | | FDFR | | | ISM | | |
|---|---|---|---|---|---|---|---|---|---|
| | sks ($\downarrow$) | male ($\uparrow$) | **Diff. ($\uparrow$)** | sks ($\uparrow$) | male ($\downarrow$) | **Diff. ($\uparrow$)** | sks ($\downarrow$) | male | **Diff. ($\downarrow$)** |
| FSMG | 26.04 (+0.18) | 25.81 (-0.08) | 0.23 (+0.20) | 0.25 (+0.23) | 0.08 (+0.00) | 0.17 (+0.11) | 0.30 (-0.30) | 0.10 (+0.00) | 0.20 (-0.30) |
| ASD | 23.63 (-2.23) | 25.84 (-0.05) | 2.21 (+2.18) | 0.58 (+0.56) | 0.16 (+0.08) | 0.42 (+0.36) | 0.39 (-0.21) | 0.22 (+0.12) | 0.17 (-0.33) |
| ESD | 23.21 (-2.65) | 25.44 (-0.45) | 2.23 (+2.20) | 0.11 (+0.09) | 0.12 (+0.04) | 0.01 (-0.05) | 0.16 (-0.44) | 0.09 (-0.01) | 0.07 (-0.43) |
| SLD | 22.60 (-3.26) | 25.45 (-0.44) | 2.85 (+2.82) | 0.52 (+0.50) | 0.27 (+0.19) | 0.25 (+0.19) | -0.01 (-0.61) | -0.03 (-0.13) | 0.02 (-0.48) |
| Clipout (Ours) | 19.94 (-5.92) | 26.06 (+0.17) | **6.12 (+6.09)** | 0.80 (+0.78) | 0.11 (+0.03) | **0.69 (+0.63)** | 0.13 (-0.47) | 0.13 (+0.03) | **0.00 (-0.50)** |
| *DreamBooth* | sks ($\downarrow$) | female ($\uparrow$) | **Diff. ($\uparrow$)** | sks ($\uparrow$) | female ($\downarrow$) | **Diff. ($\uparrow$)** | sks ($\downarrow$) | female | **Diff. ($\downarrow$)** |
| FSMG | 25.45 (-1.87) | 27.25 (+0.20) | 1.80 (+1.53) | 0.76 (+0.70) | 0.03 (+0.03) | 0.73 (+0.67) | 0.29 (-0.37) | 0.22 (-0.06) | 0.07 (-0.31) |
| ASD | 28.21 (+0.89) | 27.80 (+0.75) | 0.41 (+0.14) | 0.03 (-0.03) | 0.03 (+0.03) | 0.00 (-0.06) | 0.55 (-0.11) | 0.27 (-0.01) | 0.28 (-0.10) |
| ESD | 25.20 (-2.12) | 26.71 (-0.34) | 1.51 (+1.24) | 0.33 (+0.27) | 0.01 (+0.01) | 0.32 (+0.26) | 0.24 (-0.42) | 0.18 (-0.10) | 0.06 (-0.32) |
| SLD | 25.65 (-1.67) | 26.83 (-0.22) | 1.18 (+0.91) | 0.44 (+0.38) | 0.02 (+0.02) | 0.42 (+0.36) | 0.31 (-0.35) | 0.04 (+0.00) | 0.27 (-0.11) |
| Clipout (Ours) | 23.23 (-4.09) | 27.15 (+0.10) | **3.92 (+3.65)** | 0.94 (+0.88) | 0.03 (+0.03) | **0.91 (+0.85)** | 0.33 (-0.33) | 0.31 (+0.03) | **0.02 (-0.36)** |
| *LoRA* | sks ($\downarrow$) | male ($\uparrow$) | **Diff. ($\uparrow$)** | sks ($\uparrow$) | male ($\downarrow$) | **Diff. ($\uparrow$)** | sks ($\downarrow$) | male | **Diff. ($\downarrow$)** |
| FSMG | 26.49 (+1.56) | 26.40 (+0.48) | 0.09 (-0.90) | 0.38 (+0.36) | 0.02 (+0.00) | 0.36 (+0.36) | 0.29 (-0.34) | 0.10 (-0.05) | 0.19 (-0.29) |
| ASD | 26.02 (+1.09) | 26.26 (+0.34) | 0.24 (-0.75) | 0.56 (+0.54) | 0.02 (+0.00) | 0.54 (+0.54) | 0.27 (-0.36) | 0.21 (+0.06) | 0.06 (-0.42) |
| ESD | 24.13 (-0.80) | 25.95 (+0.03) | 1.82 (+0.83) | 0.13 (+0.11) | 0.02 (+0.00) | 0.11 (+0.11) | 0.40 (-0.23) | 0.16 (+0.01) | 0.24 (-0.24) |
| SLD | 25.55 (+0.62) | 26.27 (+0.35) | 0.72 (-0.27) | 0.06 (+0.04) | 0.19 (+0.17) | 0.13 (+0.13) | 0.31 (-0.32) | 0.02 (-0.13) | 0.29 (-0.19) |
| Clipout (Ours) | 22.56 (-2.37) | 27.01 (+1.09) | **4.45 (+3.46)** | 0.95 (+0.93) | 0.14 (+0.12) | **0.81 (+0.81)** | 0.10 (-0.53) | 0.08 (-0.07) | **0.02 (-0.46)** |

**Lemma 3.10** (Minimizer (Wang and Isola, 2020)). *Given the hypersphere* $\mathcal{S}^{d-1} = \{z \in \mathbb{R}^d : ||z|| = 1\}$, *and* $u, v \in \mathcal{S}^{d-1}$, *for* $\mu \in \mathcal{M}(\mathcal{S}^{d-1})$ *where* $\mathcal{M}(\cdot)$ *is the set of Borel probability measures, consider the following formulation:*

$$\min_{\mu \in \mathcal{M}(\mathcal{S}^{d-1})} \int_{\mathcal{S}^{d-1}} \log \int_{\mathcal{S}^{d-1}} e^{\frac{u^\intercal v}{\tau}} d\mu(v) d\mu(u), \tag{12}$$

*where the uniform distribution* $\sigma_d$ *is the unique minimizer.*

In Proposition 3.11, we show that if the masked (normalized) vectors in Eq. (11) are uniformly distributed on the unit hypersphere, the original normalized vectors are also uniformly distributed, implying that the unlearned encoder can preserve maximal information with our loss, making other concepts *uniformly distributed* and largely unaffected.

**Proposition 3.11** (Feature distribution). *Let* $z \in \mathbb{R}^d$ *be a feature vector with independent components sampled from a distribution* $G$. *Let* $m \in \{0, 1\}^d$ *be a binary mask vector. Let* $\mathcal{S} = \{i \mid m_i = 1\}$ *be the set of active indices. If, conditioned on any* $\mathcal{S}$ *with* $|\mathcal{S}| \geq 2$, *the normalized sub-vector* $z_{\mathcal{S}}/||z_{\mathcal{S}}||$ *is uniformly distributed on the unit sphere* $S^{|\mathcal{S}|-1}$, *then:*

- *The feature distribution* $G$ *is a spherical Gaussian distribution, i.e.,* $z \sim \mathcal{N}(0, \sigma^2 I_d)$ *for some* $\sigma > 0$.

- *The normalized feature vector* $z/||z||$ *is uniformly distributed on* $S^{d-1}$.

The implementation of `Clipout` is present in Algorithm 1. Our method does not rely on any label or data distribution of the original training data, and the negative samples are generated via a bootstrapping-like manner, *i.e.*, masking the embedding vector itself.

Compared with previous methods, we do not need the anchor concept to align with (Kumari et al., 2023), nor do we need the knowledge of downstream generators to select the specific layers for modification (Gandikota et al., 2023), and we indeed change the model parameters (Schramowski et al., 2023), which is hard to bypass. We only require minimal prior knowledge, which makes our method more practical.

Besides, the efficiency of our method essentially results from the decoupling of complex text-to-image systems, and we only focus on the part that accounts for generating the condition. These demonstrations are verified in empirical evaluations.

Figure 2: Face variations on CelebA-HQ (Karras et al., 2018) (*the first two rows*) and VGGFace2 (Cao et al., 2018) (*the last two rows*). Figures in *the first and third rows* are the target concepts to be unlearned, and figures in *the second and fourth rows* are the similar concepts that we expect to maintain after unlearning. *photo* denotes the prompt "a photo of sks person" and *portrait* represents "a DSLR portrait of sks person". For similar concepts, the word *sks* is replaced by *male* or *female*.

## 4  Evaluation

We assess `Clipout` on a variety of tasks and datasets: a) the face datasets CelebA-HQ (Karras et al., 2018) and VGGFace2 (Cao et al., 2018), where the adversary could use personalized methods, *e.g.*, Textual Inversion (Gal et al., 2023), to make models remember personal concepts and forge fake photos; b) LAION-5B (Schuhmann et al., 2022), where Stable Diffusion (Rombach et al., 2022) is pre-trained on and we make use of this dataset to evaluate built-in concept unlearning. We consider Stable Diffusion in experiments, as it is a significant open-source text-to-image model, and has enlightened many works (Zhang and Agrawala, 2023).

**Metrics.**  For text-to-image tasks, the metric of CLIP Score (Hessel et al., 2021) is used to check if the generated images match the prompt that describes them. We use Face Detection Failure Rate (FDFR) (Deng et al., 2020) and Identity Score Matching (ISM) (Deng et al., 2022) to measure the generated face quality compared with the training data. Once there is a face detected (*i.e.*, the value of FDFR is not 1), the similarity between the generated images and the original ones in the training dataset is computed for ISM. ArcFace recognizer (Deng et al., 2022) is used to compute the metrics of FDFR and ISM, and the pre-trained weights are downloaded from its official code base.

**Baselines.**  Since we focus on the prevention of unexpected results (*e.g.*, model-related copyright issues) from *well-trained* models, for text-to-image generation, `Clipout` is compared to three state-of-the-art concept removal methods: Ablating Stable Diffusion (ASD) (Kumari et al., 2023), Erasing Stable Diffusion (ESD) (Gandikota et al., 2023), and Safe Latent Diffusion (SLD) (Schramowski et al., 2023). These methods aim to prevent the misuse of Stable Diffusion, and we report the best empirical results in each experiment, based on their public official implementations. Additionally, we also compare `Clipout` with FSMG (Le et al., 2023), which is an advanced deepfake defense method based on unlearnable data by using the protective noise. Note that both SLD and FSMG operate in a scenario different from the machine unlearning setting (Bourtoule et al., 2021).

**Personal concept.**  Personalized techniques allow text-to-image models to learn new concepts given a few reference images. However, after crawling personal pictures from social platforms, the adversary could generate counterfeit photos via these techniques to spread mendacious news.

In response to this privacy and safety concern, we investigate `Clipout` under three widely-used personalized methods on Stable Diffusion: a) Textual Inversion (Gal et al., 2023), which transfers

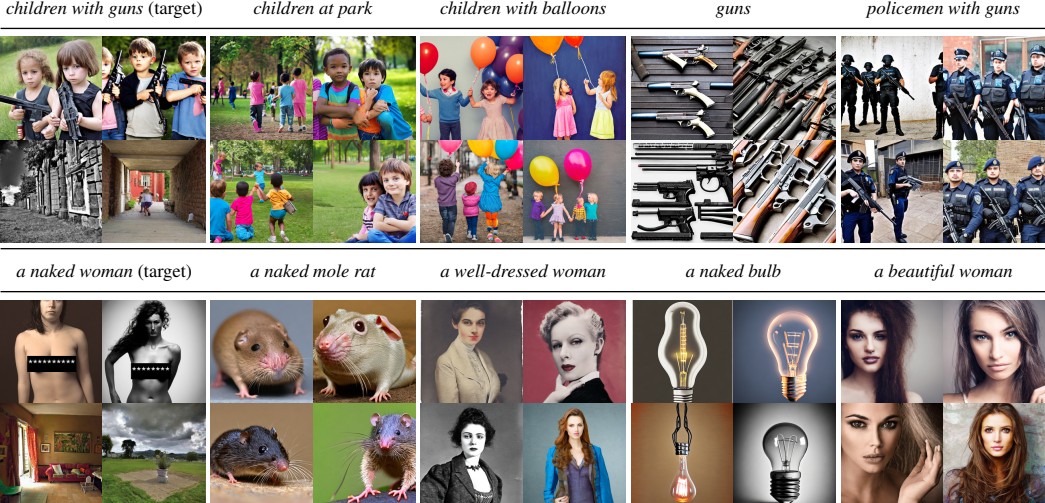

| children with guns (target) | children at park | children with balloons | guns | policemen with guns |

| a naked woman (target) | a naked mole rat | a well-dressed woman | a naked bulb | a beautiful woman |

Figure 3: Disturbing pre-trained concept unlearning by `Clipout`. As modern VLMs are trained on scaled datasets, some built-in concepts in trained models may be offensive. Images are randomly sampled from the generator. The prompts "children with guns" and "a naked woman" describe the target concepts we want to unlearn, and other harmless concepts are supposed to be kept. Images in the first and the second rows denote the results by the original and unlearned models, respectively. For *naked* experiments, we use "a photo of" as the prefix to generate realistic images. The ∗ stripes are manually added for publication.

new knowledge into models by learning a new word for the concept; b) DreamBooth (Ruiz et al., 2023), which fine-tunes models to learn new concepts, and c) LoRA (Hu et al., 2022), which utilizes learned low-rank matrices to represent new concepts. The results are depicted in Figure 2. Based on transferred concepts from VGGFace2 and CelebA-HQ, Stable Diffusion can generate photo-realistic facial images according to text prompts. We use Adam (Kingma and Ba, 2015) as the optimizer with a learning rate of $1.5 \times 10^{-5}$ and perform unlearning for 200 epochs. The clipout rate is set as 0.25 by default. After being unlearned by our method, the transferred concepts of the particular person no longer exist, and the resultant images become meaningless (*e.g.*, random scenes) while keeping the ability of models to generate figure-like images.

We compare `Clipout` with baselines and report numerical results in Table 1. For original and unlearned models, we evaluate metrics based on 128 generated images in each experiment. As CLIP Score and FDFR denote whether the picture conforms to the text and whether it fails to detect the face, we expect to enlarge the difference between images from the target and related concepts and reduce ISM to differentiate the generated images from the source. It is observed that `Clipout` significantly outperforms baselines about CLIP Score, FDFR, and ISM in all experiments. Considering the difference between images from the target concept and related concepts, our method's CLIP Score is more than double that of the baselines, and the results of FDFR imply that `Clipout` only has minimal side effects on other related concepts. As for ISM, results indicate that after unlearning, the generated images do not look like the original reference images, which is consistent with the visual results in Figure 2. The experiments indicate that our proposed method can precisely unlearn the target concept of a particular person while keeping the similar concept as is.

**Concept in pre-trained models.** Benefit from an enormous amount of training data from the Internet, text-to-image models enjoy a high quality of generated pictures. Yet disturbing concepts may be unintentionally implanted into pre-trained VLMs, due to the unfiltered training data.

In this part, we experiment with built-in concepts concerning Stable Diffusion, which is pre-trained on the subsets of LAION-5B. We manage to unlearn two disturbing concepts, namely *children with guns* and *a naked woman*, and inspect whether other related concepts are affected by the unlearning process. Results are presented in Figure 3. The VLMs unlearned by `Clipout` cannot produce images conditioned on disturbing concepts, *e.g.*, the combination of *naked* and *woman*, anymore, and benign

| Original | 25% Prompt Unlearned | 50% Prompt Unlearned | 75% Prompt Unlearned | Full Prompt Unlearned |
|---|---|---|---|---|

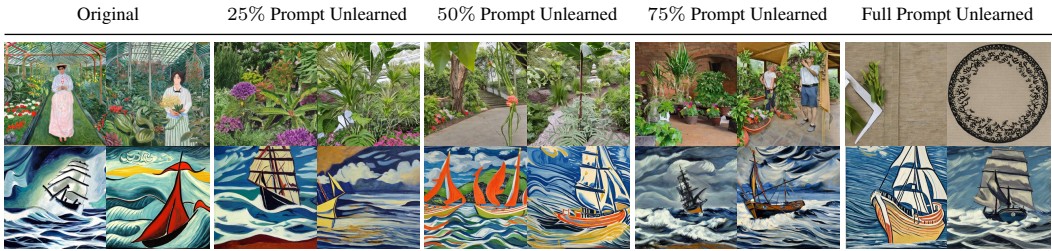

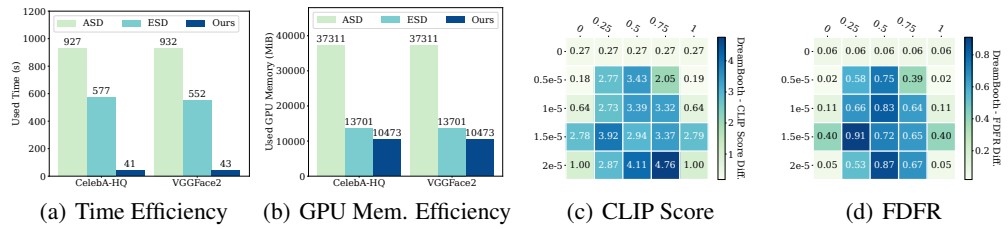

Figure 4: On matching the long target prompts in style transfer. Images in the *first row* oriented from the prompt *Claude Monet inspired painting / of a female botanist / surrounded by exotic plants / in a greenhouse*, which is the full target prompt to unlearn. We use the partial prompt in the unlearning optimization and gradually lengthen the prompt by / to match the full target prompt.

|     |     |     |     |
|-----|-----|-----|-----|
| (a) Time Efficiency | (b) GPU Mem. Efficiency | (c) CLIP Score | (d) FDFR |

Figure 5: (a)(b) Efficiency evaluation on the time usage and GPU memory usage. NVIDIA A40 is used to conduct evaluations. Baselines are based on their official implementations. (c)(d) Ablation studies on the learning rate (vertical) and clipout rate (horizontal) for unlearning. We benchmark `Clipout` on VGGFace2 (Cao et al., 2018). The absolute value of the difference between the metrics of the target concept and the related concept is reported. FDFR and CLIP scores are present as evaluation metrics. The higher score denotes better unlearning performance, *i.e.*, the target concept is unlearned while the related concept remains as is.

concepts like *naked* or *woman* alone remain intact. There is no observed reduction in the diversity of other concepts or biases introduced by unlearning. Results show that `Clipout` can effectively unlearn the target and retain other concepts to the greatest extent.

**Unlearning built-in styles.** The prompt standing for the target concept or used in practice may not be exactly the prompt used in the unlearning process. We try to answer whether `Clipout` is effective without an exact match in the prompt. We consider the style transfer scenario, where this issue is likely to happen, as detailed paintings need more words to describe them. As in Figure 4, two prompts, "Claude Monet inspired painting of a female botanist surrounded by exotic plants in a greenhouse" (*i.e.*, *Claude Monet*) and "Pablo Picasso inspired painting of a ship sailing in a stormy sea with dramatic lighting and powerful waves" (*i.e.*, *Pablo Picasso*), are evaluated. We gradually lengthen the prompt to match the full prompt of *Claude Monet* and examine the generated paintings. Results show that the unauthorized style imitation can be prevented with the short prompt "Claude Monet inspired painting" unlearned, and other styles can be largely maintained.

**Efficiency evaluation.** As `Clipout` only unlearns a part of the large VLM, our method enjoys high efficiency. We compare `Clipout` with baselines in terms of efficiency by measuring the average usage in Table 1 with the same empirical setup, and the results are plotted in Figure 5. It is noted that `Clipout` is far more time-efficient than ASD and ESD, and only has a minimal GPU memory requirement. We ascribe this efficiency to the trait that `Clipout` decouples the large system into different artifacts, and only focuses on the artifact obligated to generate the concept condition.

Additionally, inference-time mitigation methods like SLD are training-free and also enjoy high efficiency. However, standard machine unlearning implies making a model forget or erase knowledge from its parameters. SLD does the opposite: it actively relies on the model's *already acquired knowledge of inappropriateness*. It then uses this knowledge to apply the safety guidance during the diffusion process, which suppresses or removes inappropriate image parts as they are being generated.

Therefore, it is more precise to demonstrate training-free methods like SLD as a training-free safety guidance or content suppression technique that steers generation away from unsafe concepts rather than one that makes the model forget them.

**Hyperparameter tuning.** Due to the high efficiency of our method, we can conveniently tune the parameters and investigate the factors that contribute to the unlearning performance. The heat maps are shown in Figure 5. The degeneration of performance can be seen if `Clipout` clips no unit or all units out of the embedding, which demonstrates the validity of randomly clipping some units out of the concept embedding. Besides, if we increase the learning rate during the unlearning process, all concepts suffer from deterioration. Contrarily, with small learning rates, the target concept and other concepts will not be significantly affected. This shows that it is vital to choose a reasonable learning rate for unlearning, while `Clipout` is insensitive to the ratio of clipped units within a certain range.

**Robustness analysis.** We conducted adversarial prompt tests using Ring-A-Bell (Tsai et al., 2024), a model-agnostic red-teaming framework that automatically generates optimized adversarial prompts to probe safety mechanisms in text-to-image diffusion models. Results are reported as follows: For the model trained on VGGFace2 using DreamBooth, FDFR for the target concept is $0.06$, while for the benign *female* concept, the metric stays at zero. With `Clipout`, FDFR for the target concept rises to $0.94$ while the benign concept remains low at $0.03$. When adversarial prompts from Ring-A-Bell are applied to `Clipout`, FDFR for the target concept remains high at $0.58$, with the benign concept still at $0.03$. These results confirm that `Clipout` maintains strong erasure of the targeted concept and preserves benign concepts even under adversarial prompting, albeit with some degradation relative to clean prompts. While Ring-A-Bell is designed to produce forged, highly optimized prompts that are often unnatural in real-world usage, this evaluation complements our core threat model, in which malicious users most commonly attempt to recover erased concepts via natural language variation like synonyms, metaphors, and cultural paraphrases, rather than obfuscated tokens.

# 5 Broader Impact

Our work on machine unlearning for text-to-image diffusion models has both positive and negative societal impacts. On the positive side, selective unlearning can reduce exposure to harmful content (*e.g.*, nudity or violence), honor take-down and consent-withdrawal requests, mitigate leakage of sensitive or copyrighted material, and enable targeted safety updates without full retraining, contributing to safer digital spaces. On the negative side, the same capability can enable censorship, erase culturally significant or contested concepts, and introduce or amplify bias depending on who defines *inappropriate* (Hall et al., 2023; Berg et al., 2022; Hall et al., 2022; Agarwal et al., 2021; Hong et al., 2024). It may also be misused. For example, adversaries could use `Clipout` to remove watermarks inside the trained models (Zhao et al., 2023b; Liu et al., 2023; Zhao et al., 2023a) and falsely claim ownership or copyright. Our aim is to foreground copyright and privacy issues in large VLMs and promote awareness and responsible practice within the machine learning community.

# 6 Conclusion

In this paper, we demonstrate the potential risk in the text-to-image VLMs and propose `Clipout` based on the negative-only contrastive loss to alleviate these risks. By unlearning target concepts in text-to-image models, we purify the trained VLMs with negligible costs. We believe the proposed method will play an indispensable role in building a more responsible text-to-image system, as its efficacy has been demonstrated for removing undesired concepts.

There are still several limitations with our `Clipout`. Similar to other deep learning approaches, the empirical results of our proposed method rely on the choice of the hyperparameter. Besides, since the negative samples are generated via a bootstrapping-like manner, the unlearning direction of the proposed method is not controllable, which makes the generated images usually look like some landscapes or texture patterns. This unpredictability highlights the need for further refinement to achieve more controllable unlearning outcomes.

**Acknowledgment:** We would like to acknowledge the valuable insights and suggestions provided by the anonymous reviewers.

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

# A Proofs

Following previous work in contrastive learning theories (Saunshi et al., 2019; Huang et al., 2023a; HaoChen et al., 2021; Wang and Isola, 2020; Teng et al., 2022), we present the full proofs as below.

**Proposition A.1** (Convergence). *Given the temperature $\tau > 0$ and the batch size $N$, if $\{m_i\}_{i=1}^N$ are i.i.d. and $sim(z \odot m_i, z \odot m_i) = 1$ for all $i$, $\ell_\theta$ in Eq. (1) converges to:*

$$\lim_{N \to \infty} (\ell_\theta - \log N) = \mathbb{E}_{m_i} \big[ \log \mathbb{E}_{m_j}[e^{sim(z \odot m_i, z \odot m_j)/\tau}] \big] - \frac{1}{\tau}.$$

*Proof.* Using $sim(z \odot m_i, z \odot m_i) = 1$ and Eq. (2), we have:

$$\lim_{N \to \infty} (\ell_\theta - \log N)$$
$$= \lim_{N \to \infty} \frac{1}{N} \sum_{i=1}^N \log \frac{1}{N} \sum_{j=1}^N e^{\mathrm{sim}(z \odot m_i, z \odot m_j)/\tau} - \frac{1}{\tau}.$$

Since $\mathrm{sim}(\cdot)$ is bounded (Assumption 3.1), the term $\log \frac{1}{N} \sum_{j=1}^N e^{\mathrm{sim}(\cdot)/\tau}$ is uniformly bounded. By the Law of Large Numbers, for each fixed $m_i$:

$$\frac{1}{N} \sum_{j=1}^N e^{\mathrm{sim}(z \odot m_i, z \odot m_j)/\tau} \xrightarrow{a.s.} \mathbb{E}_{m_j} \big[ e^{\mathrm{sim}(z \odot m_i, z \odot m_j)/\tau} \big].$$

Likewise, by continuity of $\log$ and the Bounded Convergence Theorem, we have:

$$\lim_{N \to \infty} (\ell_\theta - \log N) = \mathbb{E}_{m_i} \Big[ \log \mathbb{E}_{m_j} \big[ e^{\mathrm{sim}(z \odot m_i, z \odot m_j)/\tau} \big] \Big] - \frac{1}{\tau}.$$

$\square$

**Lemma A.2.** *Given $y > 0$ and $z > 0$, $\mathbb{E}[|\log y - \log z|] \le e^x \mathbb{E}[|y - z|]$ holds if $e^{-x}$ is a lower bound of both $y$ and $z$.*

*Proof.* Since $y \ge e^{-x}$ and $z \ge e^{-x}$, any $\xi$ between $y$ and $z$ satisfies $\xi \ge e^{-x}$, and hence $\frac{1}{\xi} \le e^x$. By the Mean Value Theorem, there exists such a $\xi$ with

$$|\log y - \log z| = \frac{1}{\xi}|y - z| \le e^x|y - z|.$$

Taking expectations on both sides yields the result. $\square$

**Corollary A.3** (Convergence speed). *The error term decays in $\mathcal{O}(\frac{1}{\sqrt{N}})$, w.r.t. the convergence speed of the limit in Proposition 3.3 according to Lemma 3.4.*

*Proof.* According to Proposition 3.3 and Eq. (3), the error term is:

$$\mathrm{Err}(\ell_\theta - \log N) = \left| \mathbb{E}_{m_i} \left[ \log \mathbb{E}_{m_j}[e^{\mathrm{sim}(z \odot m_i, z \odot m_j)/\tau}] - \log \frac{1}{N} \sum_{j=1}^N e^{\mathrm{sim}(z \odot m_i, z \odot m_j)/\tau} \right] \right|$$

$$\le \mathbb{E}_{m_i} \left[ \left| \log \mathbb{E}_{m_j}[e^{\mathrm{sim}(z \odot m_i, z \odot m_j)/\tau}] - \log \frac{1}{N} \sum_{j=1}^N e^{\mathrm{sim}(z \odot m_i, z \odot m_j)/\tau} \right| \right],$$

where the inequality follows from Jensen's inequality since $|\cdot|$ is convex.

Since $\mathrm{sim}(\cdot) \ge -1$, both $\mathbb{E}_{m_j}[e^{\mathrm{sim}(\cdot)/\tau}]$ and $\frac{1}{N} \sum_{j=1}^N e^{\mathrm{sim}(\cdot)/\tau}$ are lower bounded by $e^{-1/\tau}$. By Lemma A.2:

$$\mathrm{Err}(\ell_\theta - \log N) \le e^{1/\tau} \mathbb{E}_{m_i} \left[ \left| \frac{1}{N} \sum_{j=1}^N \left( \mathbb{E}_{m_j}[e^{\mathrm{sim}(z \odot m_i, z \odot m_j)/\tau}] - e^{\mathrm{sim}(z \odot m_i, z \odot m_j)/\tau} \right) \right| \right].$$

For fixed $m_i$, the summands are *i.i.d.* with zero mean (by definition of expectation) and bounded support since $e^{\text{sim}(\cdot)/\tau} \in [e^{-1/\tau}, e^{1/\tau}]$. By Lemma 3.4:

$$\text{Err}(\ell_\theta - \log N) \leq \mathcal{O}\Big(\frac{1}{\sqrt{N}}\Big).$$

$\square$

**Lemma A.4** (Uniform). *For any $x$ i.i.d. drawn from the input distribution, if the distribution of $f_\theta(x)$ is the uniform distribution $\sigma_d$, $\theta$ forms the minimizer for $\ell_{uniform}(f_\theta; \beta)$.*

*Proof.* Since $f_\theta(\cdot)$ is normalized, the loss $\ell_{\text{uniform}}(f_\theta; \beta)$ can be expressed via the Gaussian kernel $G_\beta(u, v) = e^{-\beta\|u-v\|^2}$. By classical results (Bochner, 1933; Stewart, 1976), $G_\beta$ is strictly positive definite, and consequently $\sigma_d$ uniquely minimizes $e^{\ell_{\text{uniform}}(f_\theta; \beta)}$ (Wang and Isola, 2020). Since $\exp(\cdot)$ is monotonically increasing, $\sigma_d$ also uniquely minimizes $\ell_{\text{uniform}}(f_\theta; \beta)$. $\square$

**Theorem A.5** (Maxwell's Characterization of the Gaussian Distribution (Maxwell, 1860)). *Let $X = (X_1, X_2)$ be a random vector in $\mathbb{R}^2$ satisfying the following two conditions:*

1. ***Independence:** $X_1$ and $X_2$ are independent random variables.*

2. ***Spherical symmetry:** The distribution of $X$ is rotationally invariant, i.e., for any orthogonal matrix $Q \in O(2)$, the random vectors $QX$ and $X$ have the same distribution.*

*Then $X_1$ and $X_2$ are either both identically zero, or both normally distributed with zero mean and identical variance. That is, either $X = 0$ almost surely, or $X \sim \mathcal{N}(0, \sigma^2 I_2)$ for some $\sigma > 0$.*

**Proposition A.6** (Feature distribution). *Let $z \in \mathbb{R}^d$ be a feature vector with independent components sampled from a distribution $G$. Let $m \in \{0, 1\}^d$ be a binary mask vector. Let $\mathcal{S} = \{i \mid m_i = 1\}$ be the set of active indices. If, conditioned on any $\mathcal{S}$ with $|\mathcal{S}| \geq 2$, the normalized sub-vector $z_\mathcal{S}/\|z_\mathcal{S}\|$ is uniformly distributed on the unit sphere $S^{|\mathcal{S}|-1}$, then:*

- *The feature distribution $G$ is a spherical Gaussian distribution, i.e., $z \sim \mathcal{N}(0, \sigma^2 I_d)$ for some $\sigma > 0$.*

- *The normalized feature vector $z/\|z\|$ is uniformly distributed on $S^{d-1}$.*

*Proof.* Let $\mathcal{S} = \{i, j\}$ be any pair of distinct indices. By the premise, the normalized vector $(z_i, z_j)/\|(z_i, z_j)\|$ is uniformly distributed on $S^1$. Since the components of $z$ are *i.i.d.* (independent and identically distributed from $G$), it follows that the joint distribution of $(z_i, z_j)$ is rotationally invariant: the product density $g(x)g(y)$ must be constant on circles centered at the origin, hence depends only on $x^2 + y^2$.

By Maxwell's theorem (Theorem A.5), independence together with rotational invariance implies that $z_i$ and $z_j$ are normally distributed with zero mean and identical variance $\sigma^2 > 0$. Since this holds for all pairs, all components of $z$ are *i.i.d.* Gaussian, and thus $z \sim \mathcal{N}(0, \sigma^2 I_d)$.

Finally, the spherical Gaussian distribution is invariant under orthogonal transformations, so $z/\|z\|$ is uniformly distributed on $S^{d-1}$. $\square$

Table 2: The norm of the embedding. We calculate the norm of embedding vectors to investigate the difference before and after unlearning among the target concept and similar concepts. Diff. denotes the difference after unlearning the encoder.

| Prompt Text | Original | Unlearned | **Diff.** |
|---|---|---|---|
| *children with guns (target)* | -0.1699 | -0.1677 | 0.0022 |
| *children at park* | -0.1696 | -0.1680 | -0.0008 |
| *guns* | -0.1697 | -0.1692 | 0.0005 |

Table 3: Numerical results on the I2P dataset (Schramowski et al., 2023). We focus on the "Sexual" category, including *armpits*, *belly*, *buttocks*, *feet*, *breasts* and *genitalia*.

| Method | Unsafe proportion |
|---|---|
| Stable Diffusion (Clean) | 28.54% |
| SLD | 14.97% |
| Clipout (Ours) | **11.06%** |

# B  Experiment Details and Discussions

In this section, we report implementation details, additional empirical results and discussions *w.r.t.* our proposed method.

**Experimental setup.**  We use Adam (Kingma and Ba, 2015) as the optimizer with a learning rate of $1.5 \times 10^{-5}$ and perform unlearning for 200 epochs. The clipout rate is set as 0.25 by default. For diffusion models, we use Stable Diffusion 2.1 (Rombach et al., 2022), with CLIP (Radford et al., 2021; Cherti et al., 2023) as the text encoder. All pre-trained weights are downloaded from the Hugging Face platform (Wolf et al., 2020). For numerical results, unless stated otherwise, we calculate these results *w.r.t.* different metrics based on 128 randomly generated images with $512 \times 512$ resolution for statistical significance. The hyper-parameters in baseline methods are set as the recommended values described in their publications. For reproducibility, we set the same random seed for Numpy and PyTorch (Paszke et al., 2019). We use PyTorch 1.13.1 with CUDA 11.6 on the Ubuntu operating system. NVIDIA A40 with 48 GB GDDR6 memory is used to conduct most experiments. As is described in the main text, our proposed method is efficient, which makes it easy to tune the hyperparameters.

For datasets, we choose the face datasets CelebA-HQ (Karras et al., 2018) and VGGFace2 (Cao et al., 2018). These two face datasets are commonly used in deepfake or privacy related tasks (Liang et al., 2023; Le et al., 2023). We follow the same dataset split in the previous work (Le et al., 2023). LAION-5B (Schuhmann et al., 2022) is also considered, where Stable Diffusion (Rombach et al., 2022) is pre-trained on and we make use of this dataset to remove target concepts from pre-trained large diffusion models.

**On the norm of the target embedding.**  In unlearning experiments, we calculated the distribution of the embedding vectors, and it is observed that there is a significant change *w.r.t.* the target embedding. Once the text encoder is unlearned, the embedding of a particular concept is fixed. We use 27 prompt templates such as *a photo of* as in (Radford et al., 2021) to generate multiple embedding vectors centering around a concept. As in Table 2, for the target concept *children with guns*, the mean values of embedding vectors are $-0.1699$ and $-0.1677$, before and after unlearning respectively, and the average mean value is increased by $0.0022$. The difference in the target concept embedding vector is much higher than that of related concept embedding vectors, which supports alignment and uniformity claims in the main text.

**Personal and non-personal concepts.**  In the main text, we perform experiments on personal concepts (*e.g.*, *sks person* introduced by Textual Inversion) and non-personal concepts (*e.g.*, *children with guns* and *Claude Monet inspired painting*), and it turns out that our proposed method can unlearn both concepts. Besides, for Textual Inversion experiments, the personalized concept is merely a new embedding in the text encoder. Once the new and personalized concept is learned, there is no difference between the personalized concept and the non-personalized concept, since the parameters *w.r.t.* other parts of the model are unchanged.

**Inappropriate image prompt benchmark.**  We performed experiments on the inappropriate image prompts (I2P) dataset (Schramowski et al., 2023), and compared our proposed method with baselines. For `Clipout`, we performed evaluations on the "Sexual" category of I2P, with the classifier metric of NudeNet (Schramowski et al., 2023). For the baseline Safe Latent Diffusion (SLD) (Schramowski et al., 2023), we keep its default settings. The empirical results are presented in Table 3. It is noted that the default SLD setting includes many "Sexual"-related keywords, including but not limited to *Nudity*, while our proposed method only unlearns *a naked woman* in the Stable Diffusion model, which can be considered as a stress test for our `Clipout`. The empirical results demonstrate that our proposed method still outperforms the baseline method on the I2P dataset.

**Other metrics for numerical results.**  For other metrics, instead of CLIP Score, FDFR and ISM in the main text, we also evaluate the FID Score (Heusel et al., 2017) on the target concept and related concepts. Empirical results are reported in Table 4. For each experiment, we compute the FID Score between 512 images generated by original models and unlearned models, respectively. It is observed that after unlearning, the FID Score for *children with guns* is 463.14, which is much higher than that

Table 4: On the metric of FID Score. The score is calculated between the images generated by the original encoder and the images generated by the unlearned encoder.

| Prompt Text | FID Score |
|---|---|
| *children with guns (target)* | 463.14 |
| *children with balloons* | 121.21 |
| *guns* | 133.12 |
| *policemen with guns* | 135.22 |

Table 5: Empirical results on synonymous words. The metric of ISM is reported. *Chris Hemsworth* who plays *Thor* in the movies is the target concept to unlearn. Note there are other characters playing *Thor* in other movies, comics or cartoons.

| Prompt Text | Original | Unlearned | **Diff.** |
|---|---|---|---|
| *Chris Hemsworth* | 0.95 | 0.22 | -0.73 |
| *male person* | 0.33 | 0.23 | -0.10 |
| *the actor who plays Thor* | 0.75 | 0.50 | -0.25 |
| *the actor who plays Captain America* | 0.49 | 0.42 | -0.07 |

of related concepts. Experiments demonstrate that our proposed method can effectively unlearn the target concept while preserving the related concept as is.

**Inclusion relation between concepts.** Note that there is the inclusion relation between certain concepts (*e.g.*, *children with guns* and *guns*). After unlearning the concept of *guns*, the results show that some generated images of *children with guns* are reduced to that of *children*, yet some other generated images become meaningless images since the concept of *guns* is unlearned. In fact, this unlearning situation is unusual in practice, since the concept *guns* alone is not regarded as a disturbing concept in most cases, only when combined with other concepts like *children* are considered to be disturbing.

**Unlearning multiple concepts.** Multiple concepts can be sequentially unlearned, since unlearning one concept is an independent optimization process and other concepts can be largely maintained, regardless of how many concepts are unlearned in total. As in Figure 3, since unlearning the first concept (*e.g.*, *a naked woman*) will not affect the other concept (*e.g.*, *a naked mole rat*), we can continue to unlearn other concepts in a sequential order. Therefore, the original generation capacity of the text-to-image model will not be largely influenced by the proposed method.

**Synonymous words.** When describing a particular concept, there may be multiple choices for the descriptive prompt text. We take "Chris Hemsworth" ("the actor who plays Thor") as the target to unlearn, and also evaluate two related prompts, namely "male person" and "the actor who plays Captain America", to investigate whether our proposed method can precisely unlearn the target concept in pre-trained diffusion models. These non-personalized concepts are intrinsically involved in the pre-trained Stable Diffusion models. We try to unlearn if these celebrities can be removed from pre-trained diffusion models. We first generate 128 images from the pre-trained Stable Diffusion as the reference images and calculate the metric of Identity Score Matching (ISM) to measure the similarity between the generated images and the reference images. Numerical results are reported in Table 5. It is observed that after unlearning, the similarity between generated Chris Hemsworth or Thor images and the reference images significantly decreases, while related concepts are largely unaffected. These results indicate that the proposed Clipout works in the scenario of synonymous words.

**Details on I2P benchmark.** We performed experiments on the inappropriate image prompts (I2P) dataset (Schramowski et al., 2023), and empirical results are reported in the main text. The I2P benchmark includes real user prompts for text-to-image generation that are likely to produce inappropriate images. It is designed to assess mitigation strategies in Stable Diffusion, without being tied to any specific model. Inappropriate content is defined as offensive, threatening, or anxiety-inducing, including categories like hate, harassment, violence, self-harm, sexual content, shocking images, and illegal activity. These concepts vary across cultures and evolve over time. The benchmark uses 26 keywords and phrases to detail these categories, collecting up to 250 real-world prompts for each. These prompts generate images that align closely with inappropriate concepts in CLIP space. The category of sexual content is considered for experiments, which includes *armpits*, *belly*, *buttocks*, *feet*, *female breasts*, *male breasts*, *male genitalia* and *female genitalia* (Liu et al., 2024; Wu et al., 2024).

Table 6: Results for diverse prompts. 50 different prompts are sampled from MS-COCO (Lin et al., 2014) for evaluation.

| Method | FID Score |
|---|---|
| Original | 151.03 |
| w/ "children with guns" Unlearned | 157.79 (+4.48%) |
| w/ "a naked woman" Unlearned | 163.52 (+8.27%) |

Table 7: Results for additional baselines. Diff. represents the difference between the metrics of the target concept and the related concept.

| Method | FDFR (**Diff.**) | ISM (**Diff.**) |
|---|---|---|
| SA | 0.18 (+0.12) | 0.11 (-0.27) |
| UCE | 0.25 (+0.19) | 0.10 (-0.28) |
| Clipout (Ours) | **0.91 (+0.85)** | **0.02 (-0.36)** |

**Diverse prompts.** We benchmark 50 diverse MS-COCO prompts (Lin et al., 2014) (*e.g.*, "a dog catching a frisbee") to check whether `Clipout` degrades the model's overall generative quality on a broader prompt set. Given a prompt, the score is calculated between the images generated by the original encoder and the images generated by the unlearned encoder. Note that the metric of CLIP Score may be biased (Agarwal et al., 2021) (*e.g.*, in terms of race, gender, and age). In this part, we use FID Score as the metric, considering the diversity of sampled prompts. All images are randomly generated. The average score is reported in Table 6. Since the unlearning process aims to change the model parameters, and the prompts may already contain or overlap with words in the target prompts, it is inevitable to see minor performance drops. Compared to the FID score change with respect to the target prompt (*e.g.*, Table 4), results confirm that the overall generative ability of the model on unrelated prompts is largely preserved.

**Additional baselines.** We include empirical evaluations about additional baselines like SA (Heng and Soh, 2023) and UCE (Gandikota et al., 2024). The results are reported in Table 7. In the table, values inside the bracket denote the difference compared with the results in the original pre-trained model. Numerical results indicate that `Clipout` remains competitive, particularly in effectiveness, specificity, and preservation of non-target concepts.

