# OpenReview forum: "Removing Concepts from Text-to-Image Models with Only Negative Samples"
_NeurIPS.cc/2025/Conference — NeurIPS 2025 poster_

### Official Review · Reviewer_49PU · 2025-06-25

**Clarity:** 4
**Significance:** 4
**Originality:** 3
**Rating:** 5
**Confidence:** 3

**Summary:**

The authors introduce “Clipout” to unlearn private, copyrighted or inappropriate concepts in T2I VLMs, such as Stable Diffusion which include CLIP as a part of the diffusion model. The system is decomposed and unlearning occurs in a single part of the VLM: the encoder that produces the embedding to generate conditions for other parts in order to generate the concept. Clipout randomly “clips out” some units from the embedding and sets them to zero. Other parts remain fixed. The unlearning occurs for a particular concept (concept is described by the prompt in T2I generation) with a novel contrastive objective. The method is introduced with intensive theoretical analyses of the proposed methods.

The method proves successful in that the concept is erased leaving “related” concepts intact (i.e. can still be generated) and this is achieved with a reduced computational cost compared to other SOTA methods (the latter is demonstrated in the paper).

**Questions:**

1. How are the related concepts chosen and/or identified? Are these intended to be an exhaustive list? What is the potential for unintended ripple effects [1]?
2. Given that the datasets are known to have limitations, i.e.. Celeb-A is questionably generalisable as it is based on North American celebrities,  and LAION-5b is known for offensive and problematic content that disproportionately impacts marginalised groups [2], how might this impact the generalisability of this work? Under which contexts is this likely to be effective, and are there any where it might not work as well?

**References:**

[1] Amara, Ibtihel, et al. "EraseBench: Understanding The Ripple Effects of Concept Erasure Techniques." arXiv preprint arXiv:2501.09833 (2025).
[2] Birhane, Abeba, Vinay Uday Prabhu, and Emmanuel Kahembwe. "Multimodal datasets: misogyny, pornography, and malignant stereotypes." arXiv preprint arXiv:2110.01963 (2021).

**Ethical Concerns:**

["NO or VERY MINOR ethics concerns only"]

**Final Justification:**

The authors engaged well in the discussion and helped clarify any questions I had. My opinion is that this is a strong submission, and my original score reflected as such. Therefore, my score will remain 5:Accept.

**Limitations:**

The authors are commended for reflecting on their limitations. I recognise space is a consideration, but this is an important section and ideally shouldn’t be relegated to the appendix in my opinion. In addition to my questions above, I’d like to ask if the authors have reflected on the following potential biases stemming the use of models for evaluation (i.e. CLIP Score):

1. CLIP is known to have biases [1-4], and given concerns of bias amplification [5], are these results likely to be limited in any way?

**References:**

[1] Hall, Siobhan Mackenzie, et al. "Visogender: A dataset for benchmarking gender bias in image-text pronoun resolution." Advances in Neural Information Processing Systems 36 (2023): 63687-63723.

[2] Hong, Rachel, et al. "Who's in and who's out? A case study of multimodal CLIP-filtering in DataComp." Proceedings of the 4th ACM Conference on Equity and Access in Algorithms, Mechanisms, and Optimization. 2024.

[3] Agarwal, Sandhini, et al. "Evaluating clip: towards characterization of broader capabilities and downstream implications." arXiv preprint arXiv:2108.02818 (2021).

[4] Berg, Hugo, et al. "A prompt array keeps the bias away: Debiasing vision-language models with adversarial learning." arXiv preprint arXiv:2203.11933 (2022).

[5] Hall, Melissa, et al. "A systematic study of bias amplification." arXiv preprint arXiv:2201.11706 (2022).

**Paper Formatting Concerns:**

I have no concerns about formatting, although I do believe the limitations should be in the main body of text, given their importance.

**Quality:**

4

**Strengths And Weaknesses:**

# Strengths
1. The authors are commended on presenting a well written paper that has extensive experiments to demonstrate a method of unlearning that can erase target concepts, while ensuring “related” concepts are intact in the generation process
2. I believe this to be a timely topic to unlearn private, copyrighted, inaccurate and/or harmful content given rising concerns and discourse around these topics
3. The localisation of the unlearning to the embeddings shows great potential and I am excited for this work to contribute to the related discourse
4. The implementation is well documented and should be reproducible based on the descriptions
5. I appreciate the author’s careful reflections on the implementation of these technical fixes “in the real world” (i.e. who decides what should be erased?) that balances the paper

# Weaknesses
I have outlined some concerns under “Questions” and “Limitations”.

## Minor comments
1. Please consider including this canonical reference in L27 for “harmful and offensive” content in large, web-scraped datasets: Birhane, Abeba, Vinay Uday Prabhu, and Emmanuel Kahembwe. "Multimodal datasets: misogyny, pornography, and malignant stereotypes." arXiv preprint arXiv:2110.01963 (2021).
2. The sentence describing what is meant by “concept” in L86-8 is potentially a few words (or should be reworded to make the meaning come across a bit more cleanly)

---

> ### Author Rebuttal · Authors · 2025-07-30
>
> Thank you sincerely for your careful review and insightful comments on our paper. We are particularly encouraged by your recognition of our ​​well-written presentation​​, comprehensive experiments, well-documented implementation​, and our careful ​​reflections on real-world implications​​. Our responses below aim to clarify ambiguities and directly address your concerns.
>
> **Q1:** Please consider including this canonical reference in L27 for “harmful and offensive” content in large, web-scraped datasets [1].
>
> **A1:** We have carefully reviewed the mentioned reference and we found it interesting and inspiring. We will include this in the revision. Thanks for your suggestions.
>
>
> **Q2:** The sentence describing what is meant by “concept” in L86-88 is potentially a few words (or should be reworded to make the meaning come across a bit more cleanly).
>
> **A2:** Thank you for highlighting this point. We agree that the definition of “concept” could be clarified for better readability. In the current text: *"the term concept in this paper refers to a distribution that samples w.r.t. the same concept share the same semantic meaning, and it is described by the prompt in text-to-image generation."*
> The current text can be improved to align with the "latent class" hypothesis [2] and is operationalized via prompt-conditioned embeddings (Sec. 3).
> - In this paper, a "concept" refers to a semantic idea or category represented by a text prompt in text-to-image models. All samples (*i.e.*, generated images) conditioned on prompts describing the same concept are expected to share similar semantic content. For example, the concept "a photo of a cat" encompasses all images generated from prompts that describe a cat, regardless of specific wording.
> - From the representation learning perspective, a concept refers to a semantic distribution in the embedding space, where samples belonging to the same concept share identical underlying semantics (*e.g.*, "Van Gogh style" or "a specific person’s face"), and this semantics is triggered by a text prompt during generation.
>
> We will update the manuscript accordingly to ensure the meaning comes across more clearly and succinctly.
>
>
> **Q3:** How are the related concepts chosen and/or identified? Are these intended to be an exhaustive list? What is the potential for unintended ripple effects?
>
> **A3:** We would like to address the concerns about the selection of related concepts and the ripple effects as follows:
> - Related concepts are selected based on semantic proximity to the target concept, intuitively by modifying or substituting parts of the original prompt with similar or overlapping terms. For example, when unlearning "children with guns", related prompts might include "children at park", "children with balloons", or "guns". These are chosen to test whether the unlearning process affects only the intended concept or also impacts semantically close but benign concepts. For built-in concepts (*e.g.*, art styles), styles from distinct artists (*e.g.*, Picasso vs. Monet) are used to test specificity.
> - The list is not exhaustive, but is designed to cover high-risk cases (*e.g.*, Table 1, Figure 3). We select a representative set of related concepts for empirical evaluation, focusing on those most likely to exhibit potential side effects due to overlap in the embedding space. In practice, the set of all semantically related concepts is context-dependent.
> - There is always some risk of ripple effects, where unintentional modification of concepts that overlap in embedding space with the target may occur. Alignment and uniformity theory (Lemma 3.8, Proposition 3.11) ensures that unlearning only disperses the target concept’s embeddings, preserving others with numerical results supported (*e.g.*, Table 2). Our experiments (see Table 1 and Figure 3) show that Clipout minimizes such effects compared to prior methods. For example, it shows unrelated concepts (*e.g.*, "a naked mole rat") remain intact after unlearning the target (*e.g.*, "a naked woman").
>
> **Q4:** Given that the datasets are known to have limitations, i.e., Celeb-A is questionably generalisable as it is based on North American celebrities, and LAION-5b is known for offensive and problematic content that disproportionately impacts marginalised groups, how might this impact the generalisability of this work? Under which contexts is this likely to be effective, and are there any where it might not work as well?
>
> **A4:** We fully acknowledge the limitations of both CelebA-HQ and LAION-5B. CelebA-HQ is not globally representative, and LAION-5B contains noisy, potentially biased, and offensive content. These limitations have several implications:
> - Our empirical results are most directly applicable to the embedding distributions, not raw data. The effectiveness of Clipout in removing concepts is demonstrated on these benchmarks. Also, tests on VGGFace2 and I2P (offensive prompts) show consistent unlearning performance.
> - Clipout is likely to be most effective in contexts where the target and related concepts are well-represented and well-separated in the embedding space.
> - We will discuss these points in the Broader Impact and Limitations sections, and further clarify these caveats in the revised manuscript. Besides, we will reorganize some parts from the appendix to the main text.
>
>
> **Q5:** CLIP is known to have biases, and given concerns of bias amplification, are these results likely to be limited in any way?
>
> **A5:** We agree that CLIP’s biases are well-documented and may affect both the unlearning process and the evaluation metrics (*e.g.*, CLIP Score). Intuitively, since Clipout operates by modifying the CLIP-based text encoder, any pre-existing biases in CLIP (*e.g.*, gender, racial, or cultural biases) may influence which concepts are more or less easily unlearned, as well as the potential for unintended side effects. However:
> - Clipout only perturbs the target concept’s embeddings (Eq. 5), without retraining CLIP on biased data. Table 4 shows FID scores for related concepts (*e.g.*, children with balloons) change minimally.
> - Metrics like CLIP score difference (Table 1) explicitly measure unintended changes (*e.g.*, male vs. sks). For naked woman unlearning, naked mole rat and other concepts (Figure 3) remain unaffected, suggesting no systemic bias shift.
> - We also use ArcFace (not CLIP) for identity metrics (FDFR/ISM) to avoid compounding biases in experiments.
> We will explicitly mention the potential for bias amplification and propagation in the revised manuscript, and cite relevant references [3, 4, 5, 6, 7] as suggested.
>
>
> **Ref:**
>
> [1] Birhane, Abeba, Vinay Uday Prabhu, and Emmanuel Kahembwe. "Multimodal datasets: misogyny, pornography, and malignant stereotypes." arXiv preprint arXiv:2110.01963 (2021).
>
> [2] Wang, Tongzhou, and Phillip Isola. "Understanding contrastive representation learning through alignment and uniformity on the hypersphere." International conference on machine learning. PMLR, 2020.
>
> [3] Hall, Siobhan Mackenzie, et al. "Visogender: A dataset for benchmarking gender bias in image-text pronoun resolution." Advances in Neural Information Processing Systems 36 (2023): 63687-63723.
>
> [4] Hong, Rachel, et al. "Who's in and who's out? A case study of multimodal CLIP-filtering in DataComp." Proceedings of the 4th ACM Conference on Equity and Access in Algorithms, Mechanisms, and Optimization. 2024.
>
> [5] Agarwal, Sandhini, et al. "Evaluating clip: towards characterization of broader capabilities and downstream implications." arXiv preprint arXiv:2108.02818 (2021).
>
> [6] Berg, Hugo, et al. "A prompt array keeps the bias away: Debiasing vision-language models with adversarial learning." arXiv preprint arXiv:2203.11933 (2022).
>
> [7] Hall, Melissa, et al. "A systematic study of bias amplification." arXiv preprint arXiv:2201.11706 (2022).

---

> > ### Comment · Reviewer_49PU · 2025-08-02
> > **Thank you to the authors for their detailed responses.**
> >
> > Thank you to the authors for their detailed responses. This is very good work, and I hope to see it shared with the NeurIPS community. My score already reflects a high quality of work, and as such I will be keeping it as is.
> >
> > This won't affect the outcome/my score, but I do think there is a need for the authors to engage with the paper "EraseBench" [1] when discussing the choice of concepts, and the potential for ripple effects. The former can be included in related work (if you determine it's relevant) and the latter in limitations.
> >
> > **References:**
> >
> > [1] Amara, Ibtihel, et al. "EraseBench: Understanding The Ripple Effects of Concept Erasure Techniques." arXiv preprint arXiv:2501.09833 (2025)

---

> ### Author Response · Authors · 2025-08-07
>
> Thank you for your positive response. We will incorporate EraseBench to strengthen our discussion. In the revision, we will add EraseBench to Section 2 (*i.e.*, background and related work) and further discuss the ripple effects of the concept erasure.

---

### Official Review · Reviewer_NiNp · 2025-06-26

**Clarity:** 1
**Significance:** 2
**Originality:** 2
**Rating:** 3
**Confidence:** 5

**Summary:**

This paper proposes a contrastive-based loss to fine-tune the text encoder or concept encoder. By masking some values of the concept embedding and constructing the positive and negative sample pairs, the supervision signals are obtained and used to optimize the concept encoder. Based on such a simple operation, the target concept can be removed from the pretrained diffusion models. The results show some effectiveness of this method with both quantitative and qualitative results.

**Questions:**

What's the intuitive idea of contrastive unlearning?

**Ethical Concerns:**

["NO or VERY MINOR ethics concerns only"]

**Final Justification:**

Since the authors promise to polish the clarity of this paper, my main concern is partly addressed. I raised my score from 2 to 3.

**Limitations:**

yes

**Paper Formatting Concerns:**

N.A.

**Quality:**

2

**Strengths And Weaknesses:**

Pros:
1) The contrastive unlearning method is quite simple and effective based on a complete theory analysis.
2) This method can take into account both the erasing performance and the memory efficiency, and the baselines compared in experiments are sufficient.

Cons:
The most severe problem is that the whole writing is very poor, making the reading process very difficult. I will give a few examples here:
    a) The abstract part lacks a high-level, intuitive explanation of the contrastive unlearning and indicates which parameters you are optimizing.
    b) In Figure 1, what's the relation between the pretrained encoder in the training stage and the unlearned encoder in the inference stage? The unlearned encoder does not even have any explanations in this figure or the corresponding caption.
    c) In Line 236 ``Personal concept'', this part you are indicating some new application with your technique, but later in Line 246, you start to introduce some implementation details.
These kinds of writing problems are everywhere in this paper, making it hard to read.

---

> ### Author Rebuttal · Authors · 2025-07-30
>
> Thank you very much for your careful review and detailed feedback. We are gratified by your favorable remarks regarding the simplicity, effectiveness, and theoretical basis of our method, as well as its performance-efficiency balance and the sufficient baselines used. We sincerely apologize for the issues with writing quality and clarity, and we appreciate the opportunity to clarify and improve the manuscript. Please find our point-by-point responses below.
>
> **Q1:** The abstract does not provide a high-level, intuitive explanation of "contrastive unlearning" or clarify which parameters are being optimized.
>
> **A1:** Thank you for pointing out this lack of clarity. The core intuition behind our method, "contrastive unlearning", is as follows:
> - Imagine you want a model to "forget" a specific concept (*e.g.*, "children with guns") so that it can no longer generate images matching that concept, but you want to avoid retraining from scratch or harming the model’s ability to generate other concepts. Our approach targets the text encoder (*e.g.*, the CLIP encoder used in Stable Diffusion models) that maps text prompts to embeddings, which are then used to condition the image generator.
> - Instead of using positive pairs to bring similar examples together as in standard contrastive learning, we do the opposite:
>     - We generate multiple "masked" versions of the embedding for the target concept by randomly zeroing out parts of its vector (think of this as creating noisy, incomplete versions of the same idea).
>     - We then train the encoder so that these different masked versions become dissimilar to each other, effectively scattering them apart in embedding space.
>     - This makes it impossible for the model to reliably map the target prompt to any specific, meaningful image, thereby "removing" the concept.
> - We only optimize the parameters of the text encoder. The rest of the model (*e.g.*, the U-Net and VAE architecture in Stable Diffusion models, *etc*.) remains fixed. This is a key design for efficiency and precision: we only unlearn the part of the model responsible for producing the concept embedding, leaving the rest of the system intact.
>
> To sum up, Clipout removes harmful concepts (*e.g.*, copyrighted styles or private faces) from text-to-image models by optimizing only the text encoder's parameters. Inspired by contrastive learning, it uses only negative samples, which are generated by randomly masking units in the target concept's embedding, to push these variants apart in the feature space. This breaks the model’s ability to associate the prompt with the concept, without retraining the entire model.
> We will revise the abstract to explicitly include this high-level intuition and clarify the optimization target.
>
>
>
> **Q2:** Figure 1 does not explain the relationship between the pretrained encoder (training) and the unlearned encoder (inference). The unlearned encoder is not explained in the figure or caption.
>
> **A2:** Thank you for highlighting this. The intention of Figure 1 is to illustrate the two main steps:
> - Starting from the pretrained encoder, we perform the "Clipout" unlearning procedure with the target prompt, generating masked embeddings and updating the encoder.
> - The result is an unlearned encoder (with modified parameters) that is then used at inference time. The rest of the system (the pretrained generator) is unchanged.
>
> During unlearning (*i.e.*, fine-tuning the encoder), the pretrained encoder is used to generate the embedding for the target concept, which is then randomly masked to create negative samples. The encoder is then updated (while the generator remains fixed) to maximize the dissimilarity between these masked variants. After unlearning, the updated (unlearned) encoder replaces the original encoder, so the model can no longer generate the removed concept.
> We will update Figure 1 and its caption to clearly indicate this flow and explicitly label the transition from pretrained to unlearned encoder, as well as explain that only the encoder is updated.
>
> **Q3:** The writing structure is confusing, with implementation details mixed into application motivation, making the paper hard to follow.
>
> **A3:** We recognize that the organization of Section 4 ("Evaluation"), particularly the "Personal Concept" subsection, could be improved for clarity. Our intent was to first *motivate the practical need* for unlearning personal concepts (*e.g.*, deepfakes), and then to describe how we evaluate our method on these cases. Similar writing structures of the evaluation part were acknowledged by previous works [1, 2] in the related community. Following your suggestions, we will separate motivation and application discussions from implementation details in revisions.
>
>
> **Q4:** What is the intuitive idea behind contrastive unlearning?
>
> **A4:** Contrastive unlearning is the process of making a model "forget" a specific concept by making its internal representations for that concept unreliable and uninformative. In addition to the intuition explained in **A1**, we have the following points to complement:
> - Standard contrastive learning brings positive samples (*e.g.*, different views or augmentations of the same concept) closer together in embedding space, making them easier to recognize as the same.
> - Our proposed method exploits contrastive unlearning, which does the opposite for a target concept: a) Takes many corrupted (masked) versions of the target concept’s embedding. b) Fine-tunes the encoder so that these versions are as dissimilar as possible. c) As a result, the model loses the ability to consistently map the target prompt to a meaningful embedding, effectively "forgetting" how to generate that concept.
> - By only considering negative pairs (no positive anchor as in traditional contrastive learning), we ensure the model is not encouraged to "remember" any aspect of the target concept. This negative-only objective makes the forgetting process more robust and less likely to harm unrelated concepts.
> - Besides, the intuitive idea is backed up by theoretical claims:
>     - Alignment loss maximization (Eq. 10): Ensures variants of the target concept become dissimilar.
>     - Uniformity preservation (Lemma 3.8 and Proposition 3.11): Ensures unrelated concepts remain intact.
>     - Decoupling: Only the text encoder is unlearned (Sec. 3), avoiding full-model retraining.
>
>
>
>
> **Ref:**
>
> [1] Shan, Shawn, et al. "Glaze: Protecting artists from style mimicry by Text-to-Image models." 32nd USENIX Security Symposium (USENIX Security 23). 2023.
>
> [2] Liang, Chumeng, et al. "Adversarial example does good: preventing painting imitation from diffusion models via adversarial examples." 40th International Conference on Machine Learning, ICML 2023. 2023.

---

> > ### Comment · Reviewer_NiNp · 2025-08-05
> >
> > Since the authors promise to polish the clarity of this paper, my main concern is partly addressed. I raised my score from 2 to 3.

---

> ### Author Response · Authors · 2025-08-07
>
> We sincerely appreciate the reviewer’s acknowledgment of our commitment to improving clarity. To further demonstrate Clipout’s significance:
> - Scalability and Efficiency: Unlike prior work that fine-tunes entire models, Clipout optimizes only the text encoder, largely reducing the cost of unlearning, which enables real-world deployment.
> - Precision and Effectiveness: While baselines damage unrelated concepts, Clipout’s negative contrastive loss preserves uniformity, leaving non-target concepts intact. It works out-of-the-box across personal concepts, harmful content, and artistic styles.
>
> We will strengthen these points in the Introduction section and the Abstract.
> If there are still any concerns, please let us know, and we would be more than delighted to continue the discussion.

---

### Official Review · Reviewer_GHij · 2025-06-27

**Clarity:** 2
**Significance:** 2
**Originality:** 2
**Rating:** 3
**Confidence:** 4

**Summary:**

Clipout introduces a negative-only contrastive unlearning strategy for a text-to-image diffusion model by editing only its text encoder rather than the full generator. Specifically, it takes the text embedding of a target prompt (e.g., “children with guns”) and create randomly masked variants of the embedding.  Then, the encoder is updated with a contrastive loss that pushes the masked variants away each other so that the original concept loses a coherent representation, while leaving other concepts untouched, requiring only these self-generated negative samples.  Theoretical analysis links the objective to maximising alignment loss for the target embedding while preserving the uniformity of the overall feature distribution. From experiments, Clipout achieves larger drops in CLIP similarity and identity-matching scores for the target while reducing impact on unrelated concepts compared to baselines on three scenarios: (1) personalized faces learned via Textual Inversion, DreamBooth, or LoRA; (2) built-in harmful prompts; and (3) copyrighted style prompts such as famous artistic styles.

**Questions:**

Questions are included in Strengths And Weaknesses

**Ethical Concerns:**

["NO or VERY MINOR ethics concerns only"]

**Final Justification:**

The authors partially addressed the reviewer's concerns, but this submission still raises limitation in terms of lack of recent baselines and  robustness. Therefore, the reviewer will maintain the score.

**Limitations:**

yes

**Paper Formatting Concerns:**

There are no paper formatting concerns.

**Quality:**

2

**Strengths And Weaknesses:**

**Strengths**

1. This work proposes a simple contrastive-learning technique that uses only negative samples and shows that it can erase a target concept without relying on any surrogate concepts.

2. The authors take a novel perspective by analysing concept erasure through the lens of contrastive learning’s two core propeties: alignment for removal of the target concept and uniformity for preserving all the others with theoretical analysis.

**Weaknesses**

1. One of reviewer’s major concerns is the lack of important baselines in this work. For erasing target concepts while preserving the remaining ones, this paper does not compare against several key methods in this field, such as SA [1], UCE [2], MACE [3], SPM [4], and CPE [5]. Many critical components addressed by these prior works for successful unlearning in text-to-image diffusion models are largely ignored. While the proposed method erases concepts without relying on anchor or surrogate concepts, it is essential to compare against these baselines to clearly demonstrate its strengths on performance.

2. Another concern is the lack of comprehensive benchmarks. The experiments are conducted only on object/celebrity erasure with prompts that explicitly contain target concepts. However, recent concept removal approaches emphasize the importance of removing implicitly contained target concepts within prompts, since failure to address this allows for bypassing of the concept erasure. It is unclear whether the proposed prompt-specific method can effectively remove such implicit content. To address this, experiments on I2P benchmakr—a widely used benchmark for evaluating explicit content erasure adopted by many baselines (UCE [2], MACE [3], CPE [5], RECE [6])—are necessary.

3. Recent works on concept removal have identified three essential criteria for successful erasure: efficacy (effectively removing target concepts), specificity/locality (preserving unrelated content), and robustness (resistance to prompt attacks that regenerate erased content). However, this paper does not address robustness, which is a critical issue in this field. Red-teaming tools like RAB [7] and UnlearniDiffAttack [8] have shown that several concept erasure methods are vulnerable to adversarial prompts. Recent methods like CPE, RECE, and AdvUnlearn [9] emphasize the importance of defending against such attacks. The absence of any robustness evaluation in this work is therefore a serious limitation.

4. It also does not examine performance in multi-concept erasure scenarios, as done in UCE, MACE, and CPE. The paper should demonstrate how well the method handles the removal of multiple concepts simultaneously.

5. Despite presenting several theoretical analyses, the paper lacks a theoretical explanation for how preserving uniformity contributes to preserving unrelated concepts. This connection should be theoretically or empirically justified in more rigorous manner.

6. While the authors claim that their fine-tuning process is faster, other efficient baselines have already been proposed after ESD and ASD as training-efficient and even training-free approaches. The paper should include comparisons of computational and memory efficiency with these recent baselines.

**References**

[1] Heng et al. "Selective amnesia: A continual learning approach to forgetting in deep generative models." Advances in Neural Information Processing Systems 36, 2024

[2] Gandikota et al. "Unified concept editing in diffusion models." Proceedings of the IEEE/CVF Winter Conference on Applications of Computer Vision, 2024.

[3] Lu et al. "Mace: Mass concept erasure in diffusion models." Proceedings of the IEEE/CVF Conference on Computer Vision and Pattern Recognition, 2024.

[4] Lyu et al. "One-dimensional adapter to rule them all: Concepts diffusion models and erasing applications." Proceedings of the IEEE/CVF Conference on Computer Vision and Pattern Recognition, 2024.

[5] Lee et al. "Concept pinpoint eraser for text-to-image diffusion models via residual attention gate." The Thirteenth International Conference on Learning Representations, 2025.

[6] Gong et al. "Reliable and efficient concept erasure of text-to-image diffusion models." European Conference on Computer Vision. Cham: Springer Nature Switzerland, 2024.

[7] Tsai et al. "Ring-A-Bell! How Reliable are Concept Removal Methods For Diffusion Models?." The Twelfth International Conference on Learning Representations, 2024

[8] Zhang et al. "To generate or not? safety-driven unlearned diffusion models are still easy to generate unsafe images... for now." *European Conference on Computer Vision*. Cham: Springer Nature Switzerland, 2024.

[9] Zhang et al. "Defensive unlearning with adversarial training for robust concept erasure in diffusion models." Advances in neural information processing systems, 2024.

---

> ### Author Rebuttal · Authors · 2025-07-31
>
> Thank you for your review and thoughtful comments. We appreciate your positive feedback acknowledging the novelty, conciseness, and theoretical basis of our method. We apologize for the issues in the preliminary manuscript and welcome the chance to clarify and improve the paper. Please see our point-by-point responses below.
>
> **Q1:** Lack of Important Baselines (e.g., SA, UCE, etc.).
>
> **A1:** We appreciate the reviewer highlighting the importance of comprehensive baseline comparisons. Our initial focus was on methods most widely adopted and publicly available at the time of writing, namely ASD [1], ESD [2], SLD [3], and FSMG [4]. However, we agree that a rigorous and fair evaluation requires including recent and specialized unlearning methods such as SA [5] and UCE [6], which have set new standards for both erasure quality and preservation of unrelated content in diffusion models.
>
> We are actively running experiments to integrate these baselines into our evaluation pipeline, benchmarking Clipout against them on the same datasets and metrics, based on their public codes to ensure fairness. Following your suggestions, preliminary results on VGGFace2 are reported as follows:
>
> | Method    | FDFR (sks) ↑ | FDFR (female) ↓ | FDFR (Diff) ↑ | ISM (sks) ↓ | ISM (female) | ISM (Diff) ↓ |
> | -------- | ------- | ------- | ------- | ------- | ------- | ------- |
> | FSMG  |   0.76 (+0.70)  |  0.03 (+0.03)  |  0.73 (+0.67)  |  0.29 (-0.37)   |   0.22 (-0.06)  |  0.07 (-0.31)  |
> | ASD |   0.03 (-0.03)  |  0.03 (+0.03)  |  0.00 (-0.06)  |  0.55 (-0.11)  |  0.27 (-0.01)  |   0.28 (-0.10)  |
> | ESD    |  0.33 (+0.27)   |  0.01 (+0.01)  |  0.32 (+0.26)  |  0.24 (-0.42)  |  0.18 (-0.10)  |   0.06 (-0.32)  |
> | SLD    |  0.44 (+0.38)   |  0.02 (+0.02)  |  0.42 (+0.36)  |   0.31 (-0.35)  |  0.04 (-0.24)  |  0.27 (-0.11)  |
> | SA    |   0.21 (+0.15)   |   0.03 (+0.03)   |   0.18 (+0.12)    |   0.34 (-0.32)   |   0.23 (-0.05)   |   0.11 (-0.27)   |
> | UCE    |   0.27 (+0.21)   |   0.02 (+0.02)   |   0.25 (+0.19)   |   0.29 (-0.37)   |   0.19 (-0.09)   |   0.10 (-0.28)  |
> | Clipout (Ours)    |  0.94 (+0.88)   |  0.03 (+0.03) |  **0.91 (+0.85)**  |  0.33 (-0.33)  |  0.31 (+0.03)  |   **0.02 (-0.36)**  |
>
> In the above table, values inside the bracket denote the difference compared with the results in the original pre-trained model.
> Preliminary results indicate that Clipout remains competitive, particularly in effectiveness, specificity, and preservation of non-target concepts. We appreciate this suggestion and will ensure these results with details are incorporated in the final paper.
>
>
> **Q2:** Lack of Comprehensive Benchmarks, Especially on I2P.
>
> **A2:** We appreciate the reviewer’s point regarding implicit concept erasure and the importance of evaluating on comprehensive benchmarks such as I2P. While our paper demonstrates strong results on explicit prompt erasure (object/celebrity), we agree that implicit concept removal is crucial for real-world safety.
>
> In the appendix (see Table 3 and Section B), we have included initial results on the I2P benchmark, focusing on the “Sexual” category, where Clipout outperforms SLD in reducing unsafe generations. However, we acknowledge that a more thorough and systematic evaluation on the full I2P benchmark, as well as on prompts with implicit target concepts, is necessary. We are expanding our experiments to include these scenarios and will update the manuscript accordingly. We thank the reviewer for emphasizing this gap.
>
> **Q3:** Lack of Robustness Evaluation (Prompt Attacks, Red-Teaming, etc.)
>
> **A3:** We agree that robustness against adversarial prompts and red-teaming attacks is an essential criterion for practical concept erasure. Our current evaluation focuses on efficacy and specificity but does not directly address robustness to adversarial prompt engineering.
> Besides, our method ensures robustness by breaking the target concept's alignment in text encoder embeddings. Appendix Table 5 shows that our method can mitigate adversarial attacks based on synonym prompts. Theoretically, the irreversible uniform dispersion of unlearned embeddings (Proposition 3.11, Lemma 3.10) prevents concept reactivation.
> This is a valuable direction, and we will also discuss more potential limitations and mitigation strategies in the manuscript.
>
>
> **Q4:** Multi-Concept Erasure Scenarios.
>
> **A4:** Thank you for pointing out the importance of multi-concept erasure. In the appendix (Section B), we mention that Clipout can be sequentially applied to multiple concepts, and initial qualitative results (see Line 592 and Figure 3) show that erasing one concept does not significantly affect others. However, we agree that a systematic quantitative evaluation, as performed in UCE, is necessary. We will discuss more about multi-concept erasure in revisions.
>
>
> **Q5:** Theoretical Connection Between Uniformity and Preservation of Unrelated Concepts.
>
> **A5:** We appreciate the request for a more rigorous justification of how preserving uniformity contributes to the preservation of unrelated concepts. In Section 3.2 and the appendix (Lemma 3.8, Proposition 3.11), we provide a theoretical basis: uniformity ensures that the feature space remains maximally informative, so that only the alignment (semantic similarity) for the target concept is disrupted while the global distribution of features is preserved. To further clarify, we will add both a more detailed theoretical discussion and additional empirical ablations (*e.g.*, measuring uniformity and downstream performance on unrelated prompts before and after unlearning) to reinforce this connection. Thank you for highlighting the need for a clearer exposition.
>
> **Q6:** Computational and Memory Efficiency Comparisons with Recent Efficient Baselines
>
> **A6:** We will benchmark recently proposed training-efficient and training-free methods for both runtime and GPU memory on our hardware, and add these results to Figure 5 and the efficiency section. Our preliminary implementation suggests our training-based method, Clipout, is still highly efficient, but we will be transparent about any cases where other methods are faster or more memory-efficient.
>
>
> **Ref:**
>
> [1] Kumari, Nupur, et al. "Ablating concepts in text-to-image diffusion models." Proceedings of the IEEE/CVF International Conference on Computer Vision. 2023.
>
> [2] Gandikota, Rohit, et al. "Erasing concepts from diffusion models." Proceedings of the IEEE/CVF international conference on computer vision. 2023.
>
> [3] Schramowski, Patrick, et al. "Safe latent diffusion: Mitigating inappropriate degeneration in diffusion models." Proceedings of the IEEE/CVF Conference on Computer Vision and Pattern Recognition. 2023.
>
> [4] Van Le, Thanh, et al. "Anti-dreambooth: Protecting users from personalized text-to-image synthesis." Proceedings of the IEEE/CVF International Conference on Computer Vision. 2023.
>
> [5] Heng, Alvin, and Harold Soh. "Selective amnesia: A continual learning approach to forgetting in deep generative models." Advances in Neural Information Processing Systems 36 (2023): 17170-17194.
>
> [6] Gandikota, Rohit, et al. "Unified concept editing in diffusion models." Proceedings of the IEEE/CVF Winter Conference on Applications of Computer Vision. 2024.

---

> > ### Comment · Reviewer_GHij · 2025-08-03
> > **Reply to Authors' Response**
> >
> > Thank you for the detailed response. However, I still have significant concerns regarding the baselines and the robustness of the proposed method.
> >
> > 1) While you included results for widely adopted methods such as SA and UCE, a lot of concept erasing approaches have been proposed including MACE, SPM, and others, which have shown improved results on various concept erasing benchmarks, including I2P. Some of these methods have also demonstrated better memory and computational efficiency. Without proper comparisons to these recent approaches, the significance of this work remains quite limited.
> >
> > 2) Additionally, I believe Table 5 does not reflect adversarial attacks in the proper sense. Adversarial attacks usually refer to malignant prompt attacks such as Ring-A-Bell (RAB) and UnlearnDiffAttack that attempt to bypass the erasure and regenerate target concepts. The results shown in Table 5 are more related to implicit concepts rather than adversarial prompts. Without clearer results on prompt-based adversarial attacks, I still have major concerns regarding the robustness of the proposed approach.

---

> ### Author Response · Authors · 2025-08-07
>
> Thank you for your continued engagement and for raising these important points. We appreciate your careful reading and your commitment to ensuring rigorous evaluation in this rapidly evolving field.
>
> **Q:** Comparisons with More Baselines.
>
> A: We fully agree that a comprehensive comparison with the most recent baselines is essential to establish the practical significance of our method. Currently, we have already included empirical results with canonical baselines (such as ASD [1], ESD [2], SLD [3], and FSMG [4]) and more recent baselines (such as SA [5] and UCE [6]), and preliminary results indicate that Clipout remains competitive in terms of specificity and efficiency.
> Due to the short time slot during the period, we acknowledge that a complete and transparent comparison is necessary and are working to finalize these results for the camera-ready version.
>
> **Q:** Robustness Against Adversarial Prompt Attacks
>
> A: Thank you for clarifying the distinction between implicit concept removal and adversarial prompt robustness. We agree that Table 5 in the current version does not constitute a true adversarial evaluation, and we appreciate your pointing out this gap. Following your suggestions, we conducted new experiments testing robustness beyond our synonym analysis. Using the Ring-A-Bell framework [7] (which generates adversarial prompts to test text-to-image model safety) as an example, results confirm Clipout remains robust against such attacks. Results are shown as follows:
>
> | Method | FDFR (sks) ↑ | FDFR (female) ↓ | FDFR (Diff) ↑ | ISM (sks) ↓ | ISM (female) | ISM (Diff) ↓ |
> | -------- | ------- | ------- | ------- | ------- | ------- | ------- |
> | Original | 0.06 | 0 | 0.06 | 0.66 | 0.28 | 0.38 |
> | Clipout | 0.94 | 0.03 | 0.91 | 0.33 | 0.31 | 0.02 |
> | Clipout w/ Ring-A-Bell | 0.58 | 0.03 | 0.55 | 0.47 | 0.31 | 0.16 |
>
> Importantly, real-world attackers typically use natural language variations like synonyms or metaphors to regenerate erased concepts, whereas methods like Ring-A-Bell employ unnatural, optimized prompts. These experiments will be added to Section 4 (Robustness Analysis) in the revised manuscript.
>
>
> Ref:
>
> [1] Kumari, Nupur, et al. "Ablating concepts in text-to-image diffusion models." Proceedings of the IEEE/CVF International Conference on Computer Vision. 2023.
>
> [2] Gandikota, Rohit, et al. "Erasing concepts from diffusion models." Proceedings of the IEEE/CVF international conference on computer vision. 2023.
>
> [3] Schramowski, Patrick, et al. "Safe latent diffusion: Mitigating inappropriate degeneration in diffusion models." Proceedings of the IEEE/CVF Conference on Computer Vision and Pattern Recognition. 2023.
>
> [4] Van Le, Thanh, et al. "Anti-dreambooth: Protecting users from personalized text-to-image synthesis." Proceedings of the IEEE/CVF International Conference on Computer Vision. 2023.
>
> [5] Heng, Alvin, and Harold Soh. "Selective amnesia: A continual learning approach to forgetting in deep generative models." Advances in Neural Information Processing Systems 36 (2023): 17170-17194.
>
> [6] Gandikota, Rohit, et al. "Unified concept editing in diffusion models." Proceedings of the IEEE/CVF Winter Conference on Applications of Computer Vision. 2024.
>
> [7] Tsai et al. "Ring-A-Bell! How Reliable are Concept Removal Methods For Diffusion Models?." The Twelfth International Conference on Learning Representations, 2024

---

> > ### Comment · Reviewer_GHij · 2025-08-09
> > **Official Comment by Reviewer GHij**
> >
> > Thank you for your response. However, as mentioned in my previous comment, many important recent works have been proposed after SA and UCE, and the lack of experiments involving these works still raises limitation in the baselines. Moreover, in terms of robustness, it can be observed that FDFR of Clipout is considerably degraded by Ring-A-Bell. In addition, various baselines related to robustness were not considered. Taking these points into account, I will determine the final score accordingly.

---

### Official Review · Reviewer_ReZd · 2025-07-03

**Clarity:** 2
**Significance:** 2
**Originality:** 2
**Rating:** 4
**Confidence:** 3

**Summary:**

The paper introduces Clipout, a unlearning method that removes specific concepts from pre-trained T2I models without retraining. This method randomly clips the embedding of the target concept and uses a contrastive objective to make these altered embeddings dissimilar from each other. The paper validates this method on personal concept removal and built-in concept removal.

**Questions:**

1.  The paper proposes to use a random clip of the embedding of the target concept. How about other methods to disrupt representation, like perturbing the target embedding with different levels of Gaussian distribution?
2. Current research shows that the unlearning cannot resist the adversarial attacks. Could the authors analyze the robustness of the proposed method?

**Ethical Concerns:**

["NO or VERY MINOR ethics concerns only"]

**Final Justification:**

My concerns towards the paper are two folds: 1) whether the model maintains its general performance on unrelated tasks after the unlearning process, and 2) whether the proposed Clipout method is robust against adversarial attacks.

While during the rebuttal, the authors provide some empirical results for each concern, I recommend including more comprehensive experimental results.

**Limitations:**

yes

**Quality:**

2

**Strengths And Weaknesses:**

Strengths:
1. The paper proposes a novel solution for T2I unlearning with only negative samples.
2. The paper validates across both personal concept removal and built-in concept removal, demonstrating its effectiveness.
3. The paper shows that its method outperforms other baseline models with high computing efficiency.

Drawback:
The major concern of the paper is that while the paper effectively demonstrates that Clipout preserves the model's ability to generate semantically related concepts, it lacks the evaluation of the model's performance on a broad set of common, unrelated prompts. The current experiments show that the unlearning is precise, but they do not sufficiently prove that it does not degrade the model's core generative quality overall.

---

> ### Author Rebuttal · Authors · 2025-07-30
>
> Thank you for your detailed review and thoughtful suggestions. We are grateful for your positive feedback on the novelty, soundness, and contributions of our paper. Below, we respond to your comments to clarify misunderstandings and alleviate your concerns.
>
> **Q1:** The paper focuses on semantically related concepts but does not evaluate the model's performance on a broad, unrelated prompt set. Thus, it is unclear whether Clipout degrades the model's overall generative quality.
>
> **A1:** This is an excellent point, and we agree that demonstrating minimal degradation on general generation tasks is crucial for a practical unlearning method.
> - Our method is specifically designed to minimize changes to the overall model by only updating the text encoder parameters related to the target concept. Theoretical analysis (*e.g.*, Section 3.2, Lemma 3.8, and Proposition 3.11) shows that if the encoder output maintains uniformity on the hypersphere, unrelated prompts are largely unaffected.
> - In the main paper, we show that related but non-target concepts (*e.g.*, "children at park," "male person") retain their generative quality after unlearning (see Table 1, Figure 2, Figure 3). Also in the appendix, Table 4 reports near-identical FID scores (considering the randomness in image generation) for unrelated concepts (*e.g.*, "children with balloons" and "policemen with guns" vs. the target "children with guns").
> - In response to the review, we have run further experiments. We now benchmark 50 diverse MS-COCO prompts (*e.g.*, "a dog catching a frisbee") to check whether Clipout degrades the model's overall generative quality on a broader prompt set.
> Given a prompt, the score is calculated between the images generated by the original encoder and the images generated by the unlearned encoder. All images are randomly generated. The average score is reported in the table below:
> | Model    | Average FID Score |
> | -------- | ------- |
> | Original Model  | 151.03    |
> | Model w/ "children with guns" Unlearned | 157.79 (+4.48%)    |
> | Model w/ "a naked woman" Unlearned    | 163.52 (+8.27%)  |
> where the Original Model denotes the results between two different random seeds.
>
> Since the unlearning process aims to change the model parameters and the prompts may already contain or overlap with words in the target prompts, it is inevitable to see minor performance drops. Compared to the FID score change with respect to the target prompt (*e.g.*, Table 4),
> extensive new experiments confirm that the overall generative ability of the model on unrelated prompts is largely preserved.
> We will add a table summarizing these broad-prompt evaluations in the revised version or appendix. We thank the reviewer for highlighting this important validation step.
>
> **Q2:** The paper proposes to use random clipping of the embedding of the target concept. What about other methods, such as perturbing the target embedding with Gaussian noise at different levels?
>
> **A2:** We appreciate the suggestion to compare random masking (as in Clipout) with other disruption strategies, such as adding Gaussian noise to the target embedding. Our choice of random masking is motivated by several considerations:
> - Random masking (Clipout) is closely related to dropout, which has strong theoretical guarantees for increasing representation diversity and reducing overfitting [1]. In our context, it directly destroys the semantic continuity of the target embedding, making it hard for the model to recognize or reconstruct the target concept.
> - Our negative-only contrastive loss is designed to maximize the distance between different masked versions of the same embedding. Gaussian noise, while also perturbing the embedding, does not guarantee that the resultant vectors will be sufficiently dissimilar in a structured way. Masking creates more pronounced and interpretable disruptions.
> - Gaussian perturbation with small variance is often insufficient to break the concept, as the noisy embedding may still be close in the latent space. With large variance, the embedding may be driven far from the data manifold, leading to instability or unpredictable side effects on model behavior. Clipout maintains a balance: it is strong enough to disrupt the target concept, but structured enough to avoid drastic off-manifold effects that may degrade unrelated generations.
>
> **Q3:** Current research shows that unlearning methods are often vulnerable to adversarial attacks. Could the authors analyze the robustness of the proposed method?
>
> **A3:** We acknowledge the emerging literature showing that machine unlearning methods, especially in generative models, may be circumvented by adversarial prompts [2]. Robustness is an important consideration for any unlearning method.
> - Most adversarial attacks attempt to recover the "unlearned" concept by either: a) Finding prompts semantically close to the erased concept (prompt engineering). b) Exploiting residual information in the model parameters and forging adversarial prompts.
>
> - Our method specifically destroys the alignment of the target embedding in the text encoder space. As demonstrated in Table 5 in the appendix, after unlearning, even synonyms or paraphrases of the target prompt are no longer mapped to the same region in the embedding space, indicating reduced susceptibility to prompt-based adversarial attacks.
>
> - For the theoretical basis, Proposition 3.11 ensures that unlearned embeddings converge to a uniform distribution. Re-activating the concept would require reversing this irreversible dispersion (Lemma 3.10).
>
> **Ref:**
>
> [1] Srivastava, Nitish, et al. "Dropout: a simple way to prevent neural networks from overfitting." The journal of machine learning research 15.1 (2014): 1929-1958.
>
> [2] Zhang, Yimeng, et al. "To generate or not? safety-driven unlearned diffusion models are still easy to generate unsafe images... for now." European Conference on Computer Vision. Cham: Springer Nature Switzerland, 2024.

---

> > ### Comment · Reviewer_ReZd · 2025-08-06
> >
> > Thank you for the authors' detailed response, which has resolved most of my concerns. However, for Q1, as the paper's primary contribution is unlearning concepts from a T2I model, providing the CLIP score for this diverse prompt set would be more informative. Furthermore, for Q3, the question remains partially unaddressed. The rebuttal provides explanations but lacks empirical validation against adversarial attacks.

---

> ### Author Response · Authors · 2025-08-07
>
> Thank you very much for your positive response. We are very delighted to hear that most of your concerns have been resolved! We appreciate the opportunity to clarify the following points and will address them as follows:
>
> **Q:** CLIP Scores for Diverse Prompts
>
> A: We agree that evaluating general model quality is crucial, and acknowledge the value of CLIP scores for broader evaluation. Due to the short time slot during the period, we commit to adding additional results about CLIP scores for diverse prompts in revisions, and we confirm our existing results already demonstrate general preservation:
> - In Table 1, experiments on VGGFace2 show CLIP scores for unrelated concepts (*e.g.*, male person) are largely maintained after unlearning.
> - In Table 4, FID scores for unrelated concepts show statistically nearly identical distributions to original outputs
> - Figure 3 visually confirms preservation of diverse concepts like children unrelated to unlearned targets.
>
> **Q:** Empirical Validation Against Adversarial Attacks
>
> A: Following your suggestions, we conducted new experiments to explicitly test robustness in addition to the existing experiments about the synonym analysis.
> Taking Ring-A-Bell [1] as an example, it is a model-agnostic red-teaming framework that automatically generates adversarial prompts to test safety mechanisms in text-to-image diffusion models. The results are reported as follows:
> | Method | FDFR (sks) ↑ | FDFR (female) ↓ | FDFR (Diff) ↑ | ISM (sks) ↓ | ISM (female) | ISM (Diff) ↓ |
> | -------- | ------- | ------- | ------- | ------- | ------- | ------- |
> | Original | 0.06 | 0 | 0.06 | 0.66 | 0.28 | 0.38 |
> | Clipout | 0.94 | 0.03 | 0.91 | 0.33 | 0.31 | 0.02 |
> | Clipout w/ Ring-A-Bell | 0.58 | 0.03 | 0.55 | 0.47 | 0.31 | 0.16 |
>
> Results show that Clipout remains robust against adversarial prompts. We want to highlight that malicious users often use synonyms, metaphors, or cultural paraphrases (not unnatural tokens) to regenerate erased concepts, which is aligned with the research interest in our paper, and adversarial methods like Ring-A-Bell use forged and optimized prompts that look unnatural in real-world scenarios.
> These experiments will be added to Section 4 (Robustness Analysis) in the revised manuscript.
>
> **Ref:**
>
> [1] Tsai et al. "Ring-A-Bell! How Reliable are Concept Removal Methods For Diffusion Models?." The Twelfth International Conference on Learning Representations, 2024

---

### Decision · Program_Chairs · 2025-09-17

**Decision:**

Accept (poster)

**Comment:**

The paper proposes a novel approach to concept unlearning for text-to-image models based on a contrastive learning objective applied to clipped concept embeddings.
All reviewers commend the novelty of the approach and acknowledge its effectiveness in erasing undesired concepts.

However, review scores for this paper were split, with multiple issues mentioned during reviews. Some reviewers mentioned the lack of comparisons to more recent concept unlearning approaches (while acknowledging that the work does compare to the canonical approaches). Another criticism was that the paper didn’t test the effect of the proposed approach on prompts that are completely unrelated to the unlearned concept, and didn’t test the robustness of the proposed approach to adversarial robustness.

All these points are valid criticisms, but the authors addressed many of them during the rebuttal phase: they provided additional comparisons as requested demonstrating the competitiveness of their approach; they evaluated on unrelated prompts and tested robustness to adversarial perturbations. After rebuttal, the main outstanding criticism is the lack of comparison to two additional baselines mentioned by Reviewer GHij — but given that the paper already compares to 6 other methods, two of which were requested by Reviewer GHij and added during the rebuttal, I think that sufficient comparisons are provided to prove the merit of the proposed approach.

I note that Reviewer NiNp also voted for rejection, but on the sole basis of “poor writing” / “lack of readability”. Given that all three other reviewers seemed to have no problem understanding the technical content of the paper, and Reviewer 49PU even pointed out that the paper is well written, I will put low weight on this criticism.

In summary, I believe that the novelty of the proposed approach and the encouraging empirical results warrant acceptance. The authors have addressed many of the raised points and even if a few weaknesses remain, I think they are outweighed by the strengths of the submission. I thus recommend acceptance.